# Swapped and non-swapped TRAAK states co-exist in membranes at a ratio influenced by temperature

Yue Ma [1,2], Katrin Ackermann[3], Qaiser Waheed [1,2], Vincent Postis[4], Terry K. Smith [5], Bela E. Bode [3] & Christos Pliotas [1,2] ✉

The potassium two-pore domain (K2P) ion channel TRAAK is expressed in the nervous system and regulates the fast action potential in membranes. Like all K2P channels, TRAAK possesses a distinct extracellular cap, which adopts a swapped and a non-swapped conformation[1,2]. However, the proportional representation of these two species within native membranes – and the trigger or stimulus associated with this conformational transition – are unknown. Here, we utilise Pulse Dipolar EPR Spectroscopy combined with heterologous single-subunit spin-labelling and monitor the complete conformational ensemble of TRAAK's cap domain in membranes. We demonstrate the coexistence of the swapped and non-swapped states, quantify their populations within the TRAAK ensemble, and show that the swapped conformation dominates, with the ratio being influenced by temperature. Native lipid analysis shows that TRAAK selectively associates with and is activated by signalling lipids to the exclusion of membrane-dominant phosphatidylcholine lipids from its vicinity, forming a distinct microdomain. Our approach can identify the immediate lipid environment, detect and quantify cap state populations in homo-/hetero-K2P channels and link domain swapping to specific triggers.

Two-pore domain potassium (K2P) channels maintain the background leak currents for the resting membrane potential in mammalian cells. K2P channels are subject to modulation by various triggers and stimuli, such as fatty acids, lipids, chemicals, temperature, pH, tension, and anaesthetics[3–6]. They could also be regulated by other proteins, such as Piezo1/2 mechanosensitive channels and kinases, while they exhibit distinct responses to G proteins, which play a crucial role in neuronal excitability and astrocytic glutamate release[7–12]. The expression of K2P channels is widespread across vital organs such as the heart[13–15], brain[16,17], retina[18], and pancreatic duct[19], making them valuable drug targets[20,21], important in anaesthesia[22,23], and central to disorders[20,24,25].

K2P channels are structurally distinct from all other tetrameric potassium channels, and unlike them, K2Ps form dimeric structures. Further, K2Ps possess a distinct extracellular cap, which protrudes approximately 3.5 nm from the membrane and bifurcates the transport pathway of potassium. Pore-blocking venom-derived peptides, which typically bind to the external entrance of potassium channel pores, cannot block K2P channels[26,27], suggesting the cap may sterically obstruct the pore entrance to the extracellular side.

The first two K2P channel structures of the human TWIK-related arachidonic acid-stimulated potassium (TRAAK) (PDB 3UM7)[1] and the tandem of pore domains in a weak inward rectifying potassium (TWIK-1) (PDB 3UKM)[28] channel were modelled with their cap domain in a

[1]BioEmPiRe Centre for Structural Biological EPR Spectroscopy, School of Biological Sciences, Faculty of Biology, Medicine and Health, The University of Manchester, Manchester, UK. [2]Manchester Institute of Biotechnology, The University of Manchester, Manchester, UK. [3]EaStCHEM School of Chemistry, Biomedical Sciences Research Complex and Centre of Magnetic Resonance, University of St Andrews, St Andrews, UK. [4]Wellcome Centre for Anti-Infectives Research, Division of Biological Chemistry and Drug Discovery, School of Life Sciences, University of Dundee, Dundee, UK. [5]School of Biology, Biomedical Sciences Research Complex, University of St Andrews, St Andrews, UK. ✉e-mail: christos.pliotas@manchester.ac.uk

non-swapped (NS) configuration, namely with the outer transmembrane helix (TM1) interacting with the inner helix of the same subunit (Fig. 1a). Later, through the use of stabilising antibodies or the introduction of gain-of-function mutations, TRAAK's cap was solved in a swapped (S) state, that is with the outer TM1 interacting with the inner-helix of another subunit in the dimeric complex[2,17,25,29,30]. Since then, the S conformation has been reported in all 52 out of 54 × K2P channel structures solved by cryoEM or x-ray crystallography and across different K2P channels, e.g., TRAAK, TWIK-Related K+ Channel 1 (TREK1), TWIK-Related K+ Channel 2 (TREK2), TWIK-Related Acid-Sensitive K+ Channel 1 (TASK1), TWIK-Related Acid-Sensitive K+ Channel 2 (TASK2), TWIK-Related Acid-Sensitive K+ Channel 3 (TASK3) and TWIK1[1,2,17,25,28–39] (Supplementary Table 1).

Pulse Dipolar Electron Paramagnetic Resonance Spectroscopy (PDS) is a powerful technique to monitor dynamics, folding, oligomerisation and conformational ensembles in integral membrane proteins[40–54]. This ensemble method commonly involves the incorporation of paramagnetic spin labels through cysteine-based modifications on selected protein sites to determine inter-spin distance distributions[55–58]. We utilised an approach which combines Å resolution PDS distance measurements in ion channels[40,41,54,59,60] with heterologous single-subunit spin labelling (HSS-SL). This strategy, which we purposely developed here, was essential to distinguish between the two cap states of TRAAK (Fig. 1b,c). We unbiasedly monitored the complete conformational ensemble of the TRAAK cap domain. We studied human TRAAK, for which both the S and NS conformations have been reported[1,2,29] and endeavoured to explore the (in)existence of the two states in eukaryotic lipid membranes, as well as identifying triggers (or stimuli) which would hint towards the biological (ir)relevance of K2P cap swapping. We investigated the full-length channel devoid of any truncations of functional domains, such as the C-terminus, which in K2P channels acts as a polymodal sensor through interactions with free fatty acids[61], kinases[7,9] and G-proteins[10], and in TRAAK mediates axon initial segment localisation in brain excitatory neurons[62]. Further, by using detergent-free Styrene-Maleic Acid (SMA) copolymer technology[63,64] we eliminated any detergent-induced artefacts, which often occur during membrane solubilisation. We also conducted lipidomics analysis[65–67] on the parental cell plasma membranes surrounding the SMA-encapsulated TRAAK. Our native lipid membrane analysis shows that TRAAK selectively favours phosphatidylinositol (PI) and various phosphorylated -PIs; PIP1, PIP2 and PIP3 signalling lipids, previously associated with functional effects on several K2P channels[68], while the normally dominant phosphatidylcholine (PC) lipids are excluded from TRAAK's immediate lipid environment.

We demonstrate the coexistence of the S and NS cap states and quantify their proportional representation within TRAAK's channel population in membranes. We find that the S state is the dominant cap conformation, aligning with its high prevalence in previous structural reports. PDS measurements show that the proportional representation of the two species in eukaryotic plasma membranes is altered when the temperature changes. We conclude that the NS cap state in the human TRAAK (and by extrapolation a possibility in TWIK1) channels represents a sparsely represented, but biologically relevant state, in a thermo-dependent S/NS ratio. Our PDS-based approach is not limited to TRAAK, but can also detect and quantify cap states within other K2P channels and thus link cap states to physiologically relevant conditions. This is applicable to both homo and hetero K2P dimers, within the same (or different) K2P subfamilies, a process which is known to enhance K2P channel functional diversity[69,70].

## Results

### A platform for heterologous single-subunit spin labelling (HSS-SL) in K2P channels

All 54 × K2P high-resolution structures reported to date have been obtained from truncated constructs, missing the crucial C-terminus, which is involved in K2P multimodal regulation[7,9,10,61] and in TRAAK encodes an ankyrin G-binding motif that mediates periodic co-distribution with ankyrin G[62]. All these K2P constructs used previously in structural studies only produce two-thirds of the protein's length (Supplementary Table 1). Here, we heterologously expressed the full-length human TRAAK channel and performed both structural and functional analysis. We further isolated the channel in detergent-free conditions, using SMA-encapsulated lipid particles, designed to carry one channel per disc. This approach diminishes any unspecific inter-dimer distance contributions from within the same membrane disc, which could obscure our PDS experiments.

We first exchanged the five native non-disulphide-bond forming cysteine residues in human TRAAK to serine residues, creating a construct template containing only inaccessible cysteine residues (inCysTRAAK hereafter), and subsequently introduced Cys at desired sites. We retained the native Cys78, which forms a stable disulfide bond with the equivalent Cys of the second TRAAK subunit (thus not amenable to spin labelling) and is essential to stabilise the cap (Fig. 1a). We expressed the inCysTRAAK construct and evaluated its function by electrophysiology following reconstitution in giant unilamellar vesicles (GUVs) (Supplementary Fig. 1g-l). Proteoliposomes were induced to form membrane blisters, which allowed the creation of high-resistance GOhm seals between the membrane and patch pipette. This setup enabled electrophysiological recordings in the 'inside-out' configuration under voltage clamp conditions. The Nernst equilibrium potential for K+ ($E_{K+}$) in a 10-fold concentration gradient of K+ is −59 mV. The reverse potential closely matches $E_{K+}$ (Supplementary Fig. 1i, l), confirming that the channel is functional and potassium selective. To assess tension sensitivity, incremental pressure steps were applied to the patch of proteoliposomes, resulting in progressively increasing currents (Supplementary Fig. 1g). Furthermore, a single pressure step of −50 mmHg was provided while voltage steps were incremented to further examine potassium selectivity of the mechanically induced current. Both the baseline and mechanically stimulated currents displayed a reverse potential close to $E_{K+}$ (Supplementary Fig. 1h, i). Under these conditions, inCysTRAAK exhibited activation at $1.54 \pm 0.10$ -fold above the basal current level (mean ± SEM, −50 mmHg, Vh = 0 mV, n = 4 patches, Supplementary Fig. 1i). inCysTRAAK could also be activated by arachidonic acid (AA), a known activator of TRAAK, by $1.69 \pm 0.06$ -fold (mean ± SEM, Vh = 0 mV, n = 3 patches, Supplementary Fig. 1j-l). In summary, we found that inCysTRAAK is potassium selective, can be activated by AA and is responsive to membrane tension, with activities comparable to those of the wild-type TRAAK (Supplementary Fig. 1a–f, p, q).

To test whether inCysTRAAK presents any unspecific or specific labelling of the native Cys78, we incubated the purified protein with a very large excess of the spin label MTSSL ((1-Oxyl-2,2,5,5-tetramethyl-3-pyrroline-3-methyl) methanethiosulfonate)[71–73] and could not detect any signal by continuous wave (CW) EPR (Supplementary Fig. 2b). After confirming our protein template cannot be spin labelled and forms a functional channel, we performed screening of multiple sites to identify candidates for Cys insertion for MTSSL labelling (delivering the R1 side chain) and generated in silico distance distributions for multiple sites spanning across the cap (67R1, 70R1, 93R1) and the transmembrane region (144R1). For spin labelling of a single engineered site per subunit in homodimeric TRAAK, the architecture of the cap results in highly similar predicted distances for the two states. Therefore, PDS would not have distinguished between the two cap conformations (Fig. 1b, Supplementary Fig. 3). To circumvent this, we performed an in silico heterologous single-subunit spin labelling analysis, by placing two spin labels on a single TRAAK monomer, while leaving the other subunit unlabelled. Among the pair combinations tested, the 67R1/144R1 pair showed the largest distance separation between the S and NS TRAAK cap states, providing the highest resolution (Fig. 1c, Supplementary Fig. 4). The predicted difference in the most probable

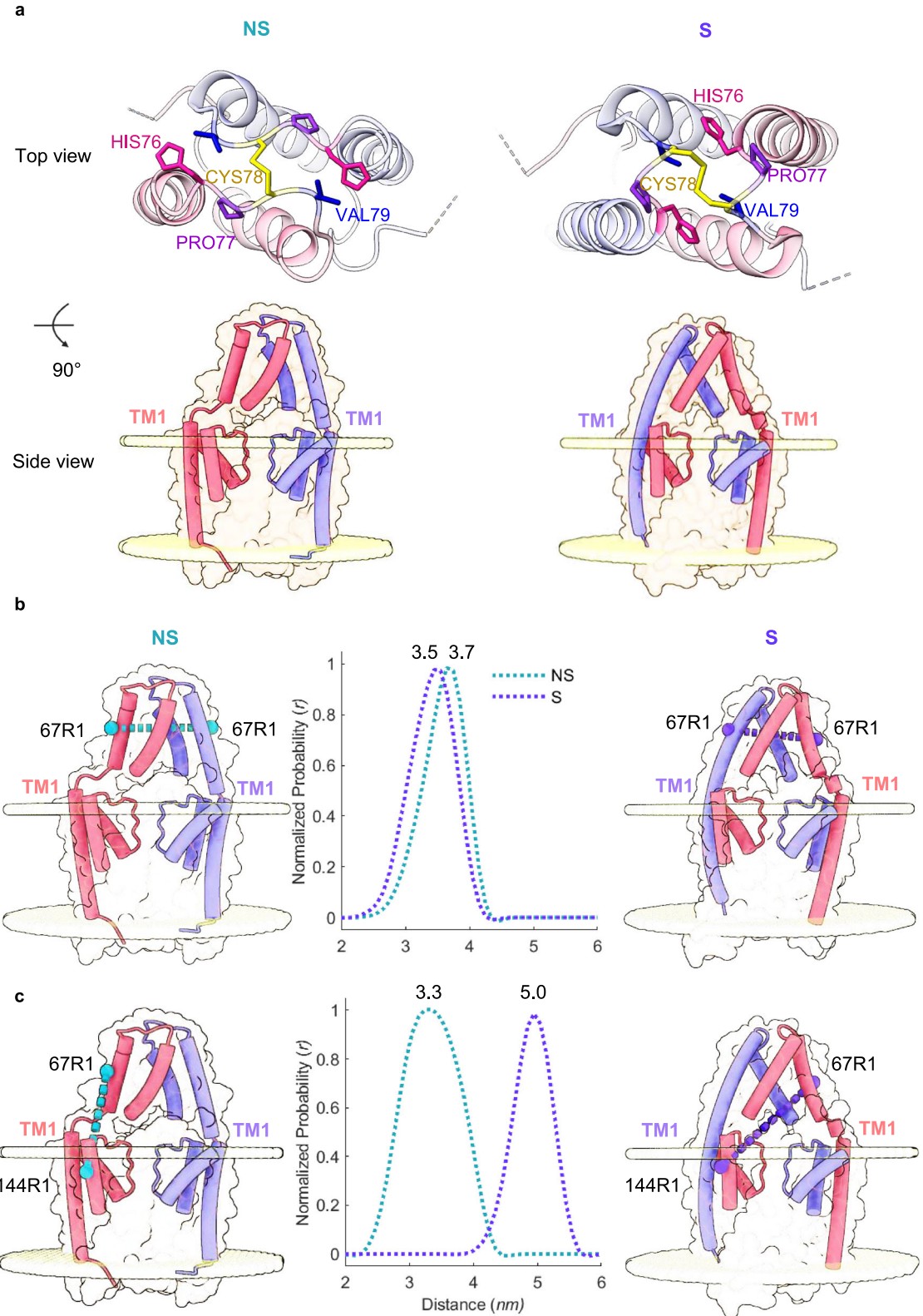

**Fig. 1 | TRAAK cap swapping and the proposed PDS-based approach to monitor the cap conformational ensemble in membranes.** Illustrations of TRAAK in the non-swapped (NS) (PDB 3UM7) and swapped (S) (PDB 4WFF) states in lipid bilayers, and in silico distance distributions of selected residues in the two states. Each subunit is depicted in a different colour (salmon or purple). **a** In the NS state, TM1 interacts with the same subunit, whereas in the S state, transmembrane helix 1 (TM1) interacts with the other subunit of dimeric TRAAK. Top-view of TRAAK showing the orientation of four residues His76, Pro77, Cys78, and Val79-differs between the NS and S states. **b** A homologous spin labelling strategy of dimeric

TRAAK cannot distinguish between the S and NS cap conformations. The spin labelled residue (67R1) in the NS (left) and S state (right). Comparison of in silico distance distributions between the two states for 67R1 is shown in the middle panel. **c** A heterologous single subunit spin labelling strategy can fully (and thus is essentially required to) distinguish between the S and NS cap conformations. Spin labelled residues 67R1 and 144R1 are depicted on only one subunit, in the NS (left) and S (right) state, with the in silico distance distributions of 67R1/144R1 for the two states shown in the middle panel. Source data are provided as a Source Data file.

distances between S and NS states is around 2 nm (mean ≈ 5 nm, SD ≈ 0.3 nm for S state; mean ≈ 3.3 nm, SD ≈ 0.5 nm for NS state) allowing unambiguous separation and quantification of the two states[74] (Fig. 1). This separation is significantly greater than twice the length of the R1 side-chain, thus beyond and excluding any biased spin label conformation that could obscure our PDS measurements. The predicted distances also fall within the optimal distance range for PDS, which is between 2 and 6 nm, allowing the two states to be distinguished.

To identify the presence of distinct cap state(s) and quantify the cap state populations within the TRAAK ensemble, we developed the HSS-SL protocol. This allowed us to ensure that one subunit of the expressed dimeric protein was spin-labelled while the other subunit remained unlabelled (i.e., EPR silent). Our protocol tailored for TRAAK could also be used for other dimeric membrane protein transporters, ion channels and pumps expressed in yeast[75]. The HSS-SL pipeline is illustrated in Fig. 2a. In a nutshell, the monomer carrying 2 × Cys (i.e., E67C/L144C) has a V5 tag and was co-transformed with the label-free inCysTRAAK, which contains a His-tag, in a 1:1 ratio. Integrated DNA sequencing confirmed that the two Cys mutants were present only in the fragment containing the V5-tag. The protocol was optimised for the EPR-active heterodimeric TRAAK population and was confirmed by Western blot analysis (Supplementary Fig. 2e). This led to three distinct isoform TRAAK species being expressed in our eukaryotic cells: a homodimer of inCysTRAAK (i.e., Cys inaccessible for spin labelling), a homodimer with two Cys pairs on each subunit (4 × Cys per dimeric channel) and a heterodimer with one Cys pair on only one of its subunits (2 × Cys per channel, i.e., our targeted species). During purification, only isoforms containing a His-tag would bind to the Ni²⁺ resin, while the homodimer, which contains 2 × Cys pairs on each subunit and carries no His-tag, would not. The EPR signal in CysTRAAK homodimer and the targeted heterodimer, which contains only one Cys-accessible pair on one of the TRAAK subunits are present in our EPR samples. Thus, only the latter species could account for the total EPR signal detected and contribute to PDS distance measurements.

Prior to initiating EPR distance measurements, we assessed the effect of E67C/L144C a) mutations and b) MTSSL modifications on TRAAK's function. To this end, we expressed, purified and reconstituted TRAAK E67C/L144C into GUVs and performed patch clamp electrophysiology under the same conditions as for inCysTRAAK (Fig. 2b–e, g, Supplementary Fig. 1m–o). We then tested whether there is an effect on channel activity, due to spin label(s) attachment (i.e., on the E67C/L144C sites) and found that MTSSL does not alter TRAAK function (Fig. 2e–g). Our recordings revealed that the modified full-length channel exhibited potassium selectivity, tension sensitivity and AA activation similar to inCysTRAAK and wild-type TRAAK (Supplementary Fig. 1p, q), showing 1.60 ± 0.04-fold activation by membrane tension over basal current levels (mean ± SEM, −50 mmHg, Vh = 0 mV, n = 4 patches. Figure 2d and 1.45 ± 0.14-fold activation by AA over basal current levels (mean ± SEM, Vh = 0 mV, n = 3 patches. Supplementary Fig. 1m–o). The reversal potential of the current was also near $E_{K+}$, indicating potassium selectivity of the modified channel (Fig. 2g)[1].

## TRAAK associates with and is activated by PI and signalling PIP lipids and excludes PC

We investigated which native lipids, originating from the parental eukaryotic plasma membrane used for expression, constitute the immediate microdomain environment of TRAAK. To this end, we performed electrospray mass spectrometry (ES-MS) analysis on the SMA-encapsulated channels to identify the lipids associated with the full-length human TRAAK. These lipids were selected by and co-residing with TRAAK immediately prior to membrane solubilisation and could hint at structural and functional roles.

We detected relatively high amounts of a range of phosphatidylinositol (PI) and signalling phosphatidylinositol phosphates (PIP)

lipids, in particular PIs (34:1, 34:2 and 34:3) and PIP, PIP₂ and PIP₃ (42:6-10) (Fig. 3a). The strikingly high amounts of PIPs found compared to their natural distribution in the plasma membrane suggests that PIPs detected here could serve as an important binding partner of full-length TRAAK. PIPs are found in the inner leaflet of the plasma membrane, waiting to be hydrolysed by phospholipase C and similar. PIPs have also been reported to bind TRAAK and modulate other K2P channels, including the TREK, THIK, TALK, and TASK subfamilies[6,68,76].

We further detected several other lipids with phosphatidylethanolamine (PE), phosphatidylserine (PS) headgroups and variable acyl-chain saturation levels. In particular, we detected PE (36:2, 36:3 and 36:4) and PS (34:1). PS is normally found in the inner-leaflet of the plasma membrane unless the cell is apoptotic, when PS is flipped to the outer-leaflet. Both PE and PS are found in human neuronal cells[77] where TRAAK natively expresses in[17] and affect the function of TRAAK[6].

Despite extensive efforts, we were unable to detect any PC lipids in the positive ion mode survey scans (Supplementary Fig. 5), even though PC is by far the most abundant lipid species in both *Pichia pastoris*[78] and human neuronal membranes[79]. This is also evident in the inositol phosphorylceramides (IPC) lipids we detected and particularly (36:2; O3 and 36:3; O3). IPC is not present in human membranes, as it is a yeast-specific lipid and is biosynthesised from PI, with both lipid species found in the plasma membrane. The equivalent of IPC in human membranes is sphingomyelin, which has a choline headgroup, and would not be present in the *Pichia pastoris* membranes, hence undetected in our analysis. Notably, the exclusion of highly abundant PC containing membranes by TRAAK is not an artefact of SMA extraction, as SMA does not have a preference for specific lipids[65], can efficiently extract PC lipids[67] and lipidomics analyses on integral membrane proteins expressed in *Pichia pastoris* and purified in SMALPs have previously detected high amounts of PC lipids[80]. Another membrane component that is surprisingly missing from our lipidomics analysis is any sterol, which would also be present at relatively high levels in the plasma membrane, so for yeast this would be ergosterol, rather than cholesterol for mammalian cells.

We then tested the effect of multiple, most abundant and representative lipids detected in our ES-MS analysis on TRAAK function using electrophysiology. We first tested each one of our patches for the presence of functional TRAAK channels by applying controlled membrane tension and subsequently incubating with variable lipids and recording channel activities. PI and PI(4,5)P₂ activated TRAAK (n = 3 patches) with ~1.5-fold, similar to arachidonic acid (AA) (Fig. 3b, c, Supplementary Fig. 1r). In contrast, DOPE (PE 36:2) inhibited TRAAK (n = 3 patches) generating a 2.00 ± 0.25-fold decrease (mean ± SEM, Vh = 0 mV, n = 3 patches; Fig. 3d). Finally, when our GUVs loaded with TRAAK, were incubated with POPS (PS 34:1) we observed no detectable effect (n = 4 patches; Fig. 3e) on TRAAK's ion current.

In summary, the lipids present are either neutral or negatively charged, PC (positive charge head-group) and sterol are distinctly excluded/absent, and the PIPs and PS are likely to be on the inner-leaflet, while the PI and PE (involved in membrane curvature) will likely be on the outer-leaflet of the membrane.

## PDS combined with HSS-SL enables probing of the entire TRAAK cap state ensemble

Prior to PDS, continuous-wave EPR (cw-EPR) confirmed that the mutated sites in each expressed protein were efficiently spin-labelled and that any excess labelling reagent was quantitatively removed (Supplementary Fig. 6). We then conducted Q-band (34 GHz) PDS distance measurements on the 67R1/144R1 pair (Fig. 4a, b) captured in native cell lipid membranes (SMALPs) at near ambient temperature (19 °C). The time traces exhibited strong visual oscillations, which led to robust distance distributions of two populations centred at 3.3 nm and 5 nm, respectively, with different weights (Fig. 4c, d). Widths and shapes of these distance populations are in full agreement with the in

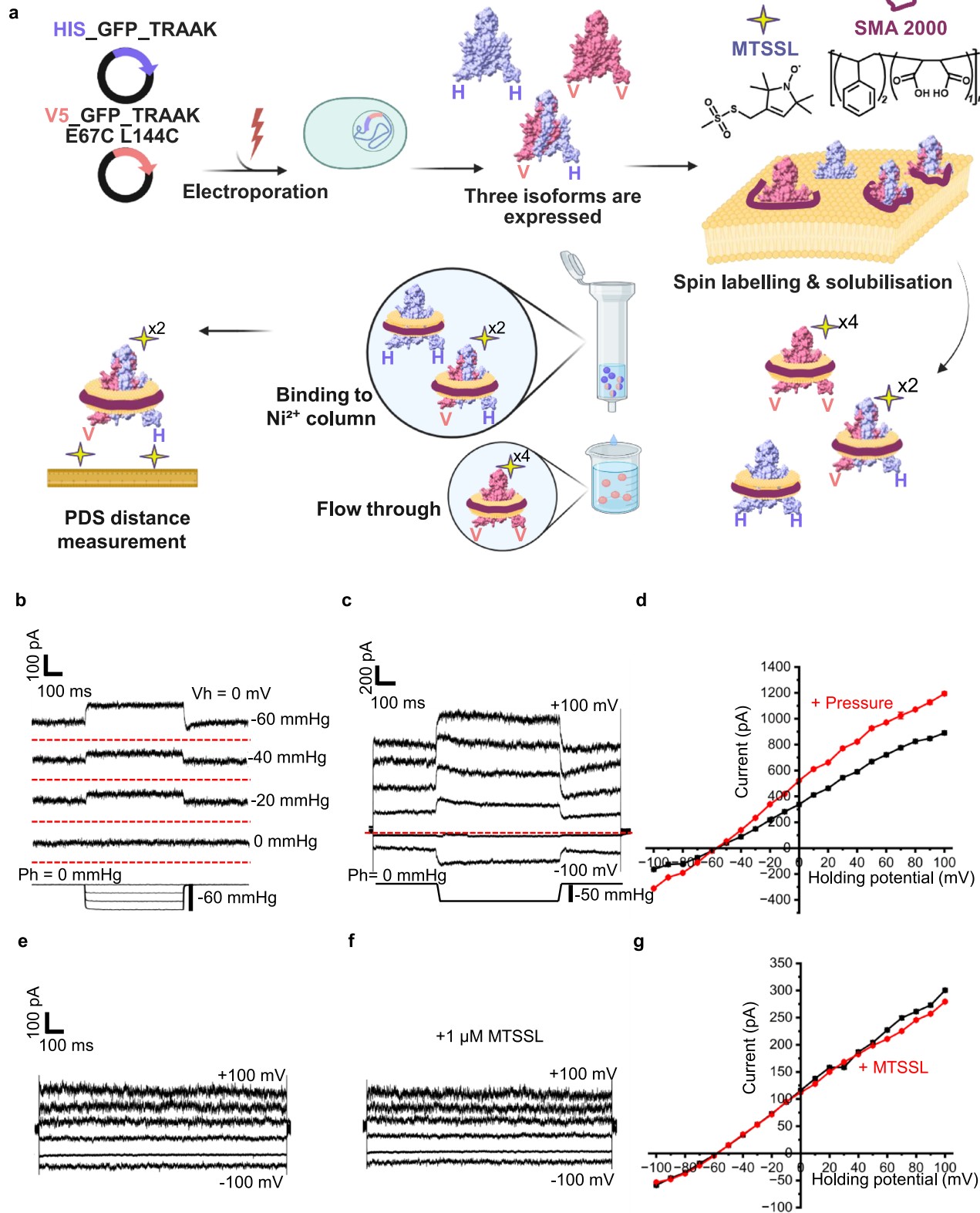

silico predictions for the cap NS and S states, suggesting both states exist within the conformational ensemble of TRAAK in the lipid membrane. These measurements were reproduced for a different protein batch, yielding almost identical PDS results (Supplementary Fig. 7). To further validate that our findings are not dependent upon a single spin pair combination[81], we performed PDS on a second heterologously spin labelled pair (70R1/93R1). For this pair, we also observed strong oscillations in the raw traces, indicating the reliability of the resulting distance distributions (Supplementary Fig. 8). However, the separation between S and NS modelled distances for this pair is not as ideal as for 67R1/144R1 (Supplementary Fig. 8d), due to limitations by the structural architecture of the K2P cap. Modelled distance distributions show significant overlap, thus the experimentally obtained trace and corresponding distance distribution cannot resolve

**Fig. 2 | Enabling the monitoring of the cap state ensemble in membranes by heterologous single-subunit spin labelling (HSS-SL). a** The HSS-SL pipeline. Co-transformation of inCysTRAAK carrying a His-tag and 2×CysTRAAK monomer carrying a V5-tag results in the expression and folding of three TRAAK isoforms: a homodimer of inCysTRAAK with a His-tag on each subunit (purple), a homodimer with one Cys pair and a V5-tag on each subunit (salmon), and a heterodimer with one Cys pair and a V5-tag on one subunit, while the other subunit is inCysTRAAK, which carries a His-tag (a mixture of purple and salmon). "H" represents the His-tag and "V" the V5-tag at the end of each monomer. Proteins are purified using detergent-free (SMA) co-polymer technology in cell membranes and spin-labelled with MTSSL. Only the EPR-silent homodimer and the targeted 2×Cys heterodimer (with one Cys-tagged subunit) from the SMA-solubilized protein mixture bind to the $Ni^{2+}$-NTA column. Proteins eluted from the $Ni^{2+}$-NTA column are further purified and subjected to PDS distance measurements, with the EPR signal originating only from the heterodimeric TRAAK population within those samples. Created in BioRender. Pliotas, C. (2026) https://BioRender.com/zx4jdu3. **b** Representative current response to pressure increments applied to the patch excised from TRAAK E67C/L144C GUVs. Similar results were observed in 3 independent biological replicates. Pressure steps were incremented every 5 sec, and the currents measured during each step are vertically offset for display. Dashed red lines below each current trace denote the zero current baseline. Holding potential (Vh) = 0 mV, holding pressure (Ph) = 0 mmHg. **c**, Representative recorded currents (Upper) obtained from a patch extracted from TRAAK E67C/L144C GUVs during a voltage step protocol (Vh = −50 mV, ranging from −100 to +100 mV, with a ΔV of 10 mV, displayed at intervals of 40 mV). For each voltage step, a pressure step of −50 mmHg was applied through the pipette (Lower). The dashed red line indicates the current level at the holding potential of −50 mV and holding pressure of 0 mmHg. Similar results were observed in 4 independent biological replicates. **d** The current-voltage relationship of the data presented in **c** is shown at 10 mV intervals. The average current prior to pressure step and the peak current during the pressure step application are plotted against each voltage. Currents from excised patches of TRAAK E67C/L144C GUVs under a voltage step protocol (Vh = 0 mV; −100 to +100 mV in 10 mV increments; every 40 mV step shown), before (**e**) and after (**f**) perfusion with 1 μM MTSSL. Similar results were observed in 3 independent biological replicates. **g** Current–voltage relationship from the data shown in (**e**) and (**f**), incremented by 10 mV. The black trace represents the average current before MTSSL application, and the red trace (+MTSSL) represents the current after perfusion with 1 μM MTSSL. Source data are provided as a Source Data file.

the presence of two distinct distance components. Nevertheless, the width of the observed distribution almost perfectly matches the combined width of the two modelled distributions, while the presence of only one of the states should have yielded a substantially narrower distribution with a shift in distance for the maximum probability. These data further support the presence of both S and NS species in the TRAAK ensemble.

To ensure that only intramolecular TRAAK distances contribute to the resulting distance distributions for both of our HSS-SL pairs and avoid any intermolecular dipolar couplings arising from adjacently localised TRAAK channels, we designed certain features and implemented experimental controls as follows.

We used an SMA type which forms lipid discs ranging from 10 to 16 nm in diameter, depending upon the lipid environment and the presence of proteins[82,83]. Such adaptability of SMA is crucial for maintaining the functional activity of TRAAK in the membrane. The TRAAK's cross-sectional TM region being approximately 8 nm, means only one TRAAK channel could fit in each lipid disc. We performed PDS for all four homo-dimeric single Cys TRAAK mutants for all the individual sites included in our heterodimeric pairs (i.e., 67R1, 70R1, 93R1, L144R1) to demonstrate efficient formation of TRAAK homodimers and rule out intermolecular distance contributions. We subsequently modified different variants of homodimeric TRAAK with MTSSL and performed PDS distance measurements in SMA (Supplementary Figs. 9, 10). For all single Cys homodimeric TRAAK mutants tested, we observed strong dipolar coupling interactions between the singly labelled homo-dimeric channels and obtained reliable distance distributions. Distances are highly consistent with the in silico predictions for the two species, which are expected to overlap, though with different mean values for the different sites and with no additional distance contributions present. We also designed single Cys hetero-TRAAK constructs (67C and 144C), but despite multiple attempts, their production has not been possible. However, any potential intermolecular distances appearing in there should also be present in our single-Cys homodimeric TRAAK distributions, but this was not the case (Supplementary Fig. 9, 10). We further constructed a 67/144 homo-dimer containing four single Cys within a single TRAAK channel to ensure that the PDS signal originates only from a single subunit, with no contribution from any other homo-oligomeric formation within the E67/L144 sample. Once again, we observed clear oscillations in the raw data of doubly homo-labelled TRAAK, which were very different from the ones recorded for hetero-labelled (67R1/144R1) TRAAK (Supplementary Fig. 11, 12).

Our multi-layered controls taken together demonstrate the absence of any intermolecular or unwanted intramolecular distances contributing to our heterologously labelled TRAAK distributions.

## NS and S states co-exist in membranes and their ratio is influenced by temperature

The experimental PDS distance distributions are in excellent agreement with predictions for the NS and S cap states (Fig. 4, Supplementary Fig. 7, 8). For quantification of populations, the shape of the distance distribution must be reliable (highest reliability range, green colour bars in Fig. 4d, f)[81]. To increase the reliability of the distance distributions, we combined short time traces with high time resolution and excellent signal-to-noise with longer time traces and analysed them globally[84,85]. While usually two traces are sufficient, we decided to include all data available (up to 4 traces) to avoid bias by selecting certain subsets of data. Despite multiple different analysis workflows and controls consistently reproducing these two peaks in the non-parametric distributions, the peak representing the S state lies outside the range where the shape of the distribution can be directly interpreted (this being ultimately limited by PDS trace length and thus echo lifetime). To overcome this technical limitation, we fitted these non-parametric distance distributions with a bimodal Gaussian model. While the quantification has inherently higher uncertainty for longer distances, we found highly consistent results using technical and biological replicates. Estimating the relative populations of NS and S cap states, we found that the majority of TRAAK is in the S state, with a substantial minority population being in the NS state S/NS: (57 ± 4 and 43 ± 4)% (with the uncertainty representing 95% confidence or 2σ). The dominance of the S state within the TRAAK ensemble aligns with the overwhelmingly frequent occurrence in the previous x-ray crystallography and cryoEM structures of TRAAK and other K2P channels. Our findings are also consistent with previous computational analysis suggesting the S state is energetically favourable[86], thereby more stable and amenable to structural studies.

Following quantification of the cap state ensemble, we next wondered whether the S/NS ratio could be influenced by any physiologically relevant triggers or stimuli. Previously, TRAAK's activity has been shown to increase by ~10-fold with a temperature increase from 17 to ~40°C[87]. We thus decided to express TRAAK under identical conditions; we incubated the membranes with SMA at two distinct temperatures, 19 °C and 40 °C. We then performed PDS on the TRAAK heterodimer 67R1/144R1 and found that at 40 °C there was a substantial change in the relative ratio between the two cap states with a

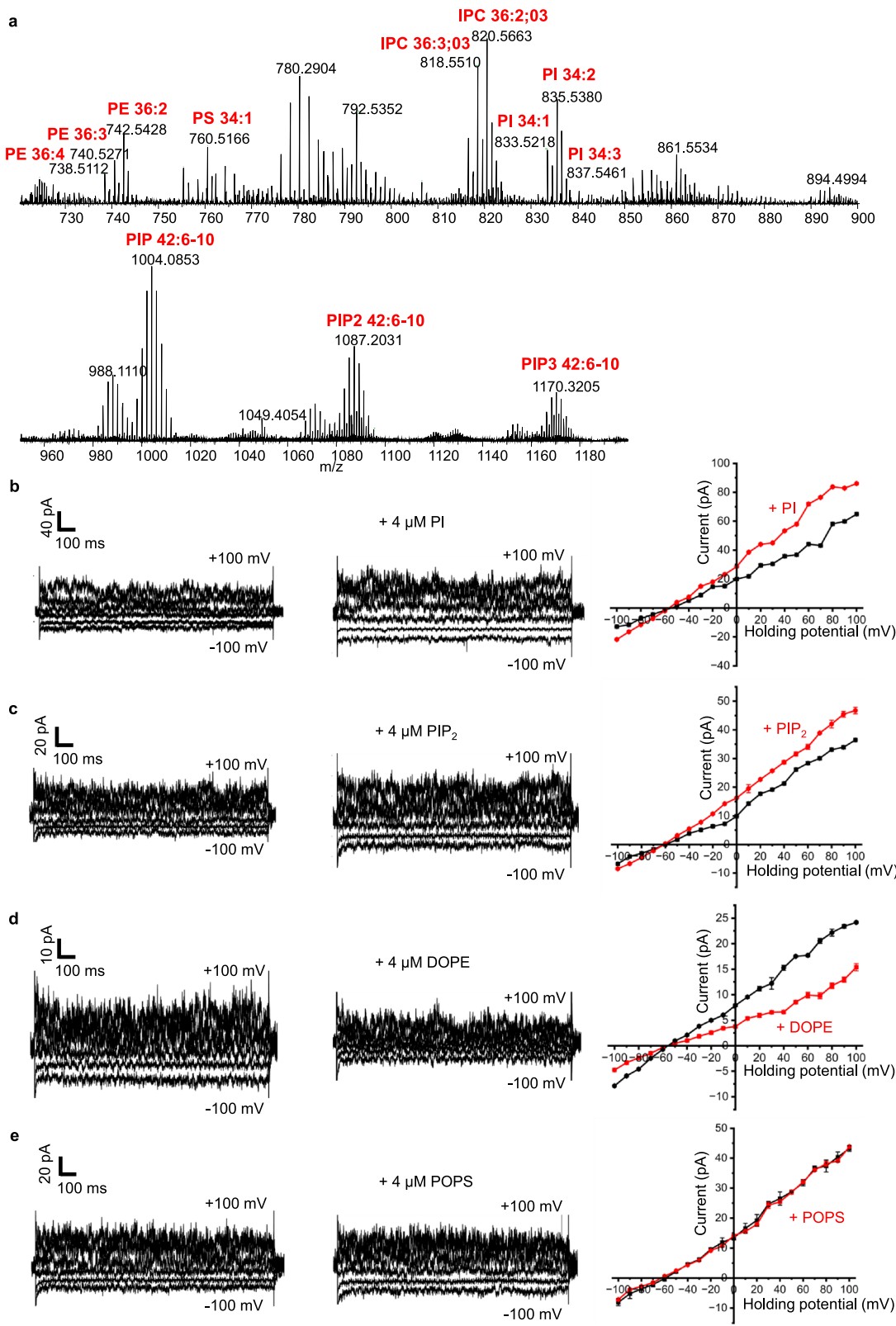

large decrease in the NS state, meaning that it becomes even rarer upon temperature increase (Fig. 4e, f). In particular, at 40 °C, the S state's dominance becomes significantly greater S/NS (66 ± 3/34 ± 3) % *vs* (57 ± 4/43 ± 4) % at 19 °C ambient temperature, with 95% confidence interval. The initial and subsequent enhancement of this dominance of the S state within the TRAAK ensemble upon temperature increase suggests the S state has greater thermodynamic stability than the NS

state. This could explain the overwhelming prevalence of the S state in previous structural reports for TRAAK in particular, and by expansion to TWIK1 and possibly other K2P channels.

To explore the effect of temperature on channel activity, we recorded electrophysiology on TRAAK reconstituted in our GUVs' setup at 24 °C and 37 °C. Considering that cap domain swapping may be a multi-step, slow process, we monitored TRAAK's activity for over

**Fig. 3 | TRAAK associates with and is modulated by specific membrane lipids. a** Native membrane lipid association of TRAAK. Lipidomics analysis of parental cell plasma membranes following SMALPs-encapsulation of TRAAK. Negative survey scans top panel (720-900 m/z); bottom panel (950-1200 m/z). The lipids identified are phosphatidylethanolamine (PE), phosphatidylserine (PS), inositol phosphorylceramides (IPC), phosphatidylinositol (PI) and phosphatidylinositol-phosphates (PIP). Numbers (x:n) represent the number of carbons: number or range of double bonds respectively, while the "On" refers to the number of oxidised double bond. Many of the detected lipids are found at low concentrations in parental yeast membranes used for expression and native neuronal plasma membranes. This applies to PI and also different types of PIP lipids involved in signalling and K2P

channel activation. In contrast, the most abundant lipid PC is not detected, suggesting TRAAK excludes this lipid from its vicinity. **b**–**e** Representative currents from excised patches of wild-type TRAAK reconstituted in GUVs during a voltage-step protocol (Vh = 0 mV; steps from −100 to +100 mV in 10 mV increments; every 40 mV step shown). Left and middle panels show recordings before and after bath perfusion with 4 μM of the indicated lipid: PI (**b**), PIP₂ (**c**), DOPE (**d**), and POPS (**e**). Right panels: I−V relationships from the same patches at 10-mV increments. Black traces indicate average currents prior to lipid application, while red traces (+PI, +PIP₂, +DOPE and +POPS) show the response following lipid perfusion. Similar results were observed in 3 independent biological replicates for each lipid. Source data are provided as a Source Data file.

1 hour at 37 °C and for shorter time periods at higher temperatures, due to patches being fragile and rupturing. We did not observe any channel activation at increasing temperatures over 37 °C (Supplementary Fig. 13). This is consistent with previous functional studies showing that TRAAK activation upon heating only occurs in a whole-cell configuration, but not in inside-out patches, suggesting that cell integrity is required for temperature sensitivity of TRAAK[5,88].

To better understand the shift of the equilibrium towards the S state, we performed independent atomistic molecular dynamics simulations of the NS state and the S state of TRAAK embedded in lipid bilayers at 19 °C and 40 °C and calculated the energies between the channel and its entire environment (Supplementary Fig. 14a, 15, Supplementary Table 4). We first ensured systems in all four conditions are in equilibrium by monitoring the RMSD of the proteins through the trajectories (Supplementary Fig. 14b). This approach enabled a direct comparison of the non-bonded interaction energy between the protein (accounting for its distinct structural architecture) and its environment (lipid bilayer and solvent) at two distinct states and different temperatures. For the S state, the temperature dependence of the protein−environment interaction is relatively small ($\Delta\Delta G \approx -979 \pm 724$ kcal mol⁻¹) and more favourable at 19 °C. The NS state exhibits a larger energy difference ($\Delta\Delta G \approx -2689 \pm 819$ kcal mol⁻¹), also favouring 19 °C. Considering the fluctuations in the energy profiles and the resulting large errors (Supplementary Fig. 14a, Supplementary Table 4), S seems to be the most stable and the least temperature-sensitive state, whereas NS becomes less stable at elevated temperatures.

To further dissect the origin of the temperature-dependent interaction energies, we analysed individual membrane properties. For the NS state, the bilayer shows larger thickness fluctuations at 19 °C than at 40 °C (Supplementary Fig. 16), indicating stronger local deformations that allow better adaptation to the protein and more favourable protein−membrane interactions at the lower temperature. In contrast, thickness fluctuations in the S state appear similar at both temperatures. We also examined the distribution of negatively charged PS lipids in the inner leaflet for all trajectories (Supplementary Fig. 17). For the NS state, PS is enriched near the channel at 19 °C compared to 40 °C. This is evident in the extended PS-rich patches of the 2D maps and the higher value in the radial distribution function g(r) of PS around the protein (Supplementary Fig. 17). For the S state, the difference in g(r) between the two temperatures is small, indicating that the PS environment is weakly dependent on temperature (Supplementary Fig. 17).

Taken together, these results suggest that the stronger temperature sensitivity of the NS state arises from a combination of factors, including temperature-dependent PS (re)organisation around the channel and changes in local bilayer mechanics.

## Discussion

The exclusive occurrence of the NS state in the first two K2P structures requires clarity regarding its biological relevance. Indeed, the proportional representation in eukaryotic plasma membranes of the S and

NS species remains unknown, as well as the triggers and/or stimuli that could induce such a channel transition to reveal its physiological role. To date, 54 × K2P structures have been solved by cryoEM or x-ray crystallography, which favour highly populated states[89]. The NS conformation's presence was not clear in cases where the electron density in the helical cap region was limited by local resolution. Given the two protomers are closely positioned in both conformations, it was challenging (and in most cases impossible) to distinguish between the two states. In many cases, the S conformation in TRAAK was obtained using antibodies to stabilise the cap (Supplementary Table 1), unavoidably leading to the selection of specific states[2,17], while attempts to crosslink the TRAAK subunits within the cell membrane led to ambiguous results[29].

Here, we purposely developed an orthogonal EPR-based method and combined it with heterologous spin labelling strategies to provide clarity. PDS is a sensitive ensemble-based technique, combined with HSS-SL, that enabled us to measure a distance separation of around 2 nm between the two cap states. Our approach allows for the detection of low-abundance states and can accurately report shifts in populations of such species as small as 9%, and link those states to specific triggers. We found that under ambient temperature, the dominant state is S (57/43), consistent with its frequent observation in previous structural reports. Despite this, the functional role of the K2P cap domain swapping remains very poorly understood and its contribution to TRAAK's multimodality has neither been considered nor excluded.

Using SMA, we captured and analysed the complete TRAAK conformational ensemble following its expression and folding into the membrane. By incubating the parental membranes at two different temperatures separated by over 20 °C, we observed significant shifts in the relative ratio between the S and NS states. These shifts may result from subunit exchange during partial unfolding and refolding or a major conformational change. This is consistent with recent observations in TREK2, where one of the two TM1 helices undergoes partial, asymmetric unfolding[39]. Both TRAAK and TREK1 have been solved in asymmetric TM4 (up/down) states, with the mid-point of one of the two TM1 helices in previous TRAAK and TREK2 structures found to be partially unwound, generating another asymmetry between channel subunits. These asymmetries suggest that two individual K2P subunits are not tightly coupled and can move independently[2,31], a requirement for facilitating a potential cap swapping transition. Such structural flexibility imposed by partial unfolding likely reduces the energy barrier required for such a major conformational change to occur, therefore facilitating transitions between the S and NS states. Our MD analysis and energy calculations qualitatively align with the equilibrium shifts in TRAAK's cap conformational ensemble informed by PDS and point towards the possibility that the observed population shifts upon heating could arise from the destabilization of the NS conformation.

A notable distinction between the S and NS states appears in the upper part of TRAAK's cap region, where both the orientation of Cys78 -essential for disulfide bond formation- and that of the three

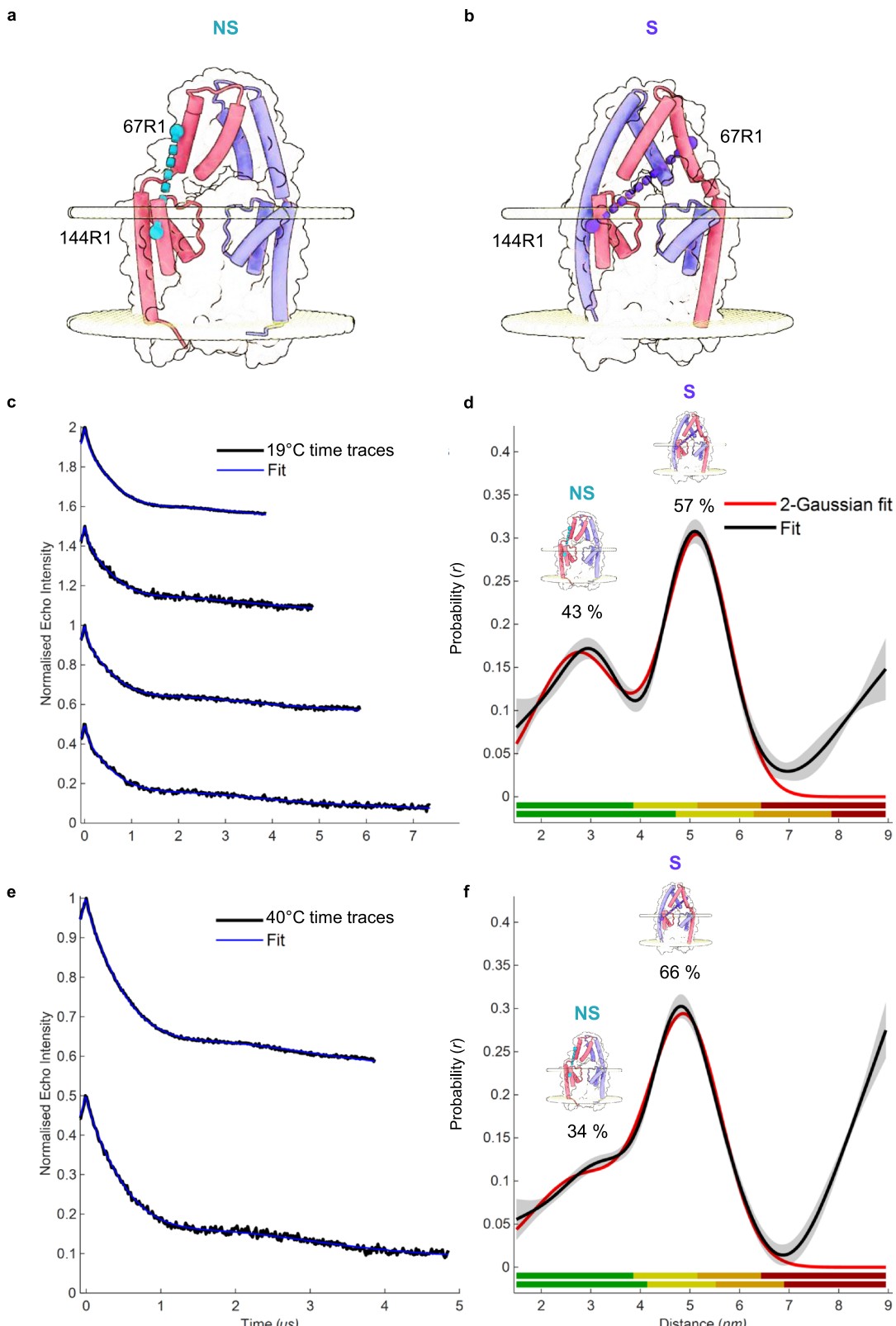

surrounding amino acids differ (Fig. 1a)[29]. Since the two states co-exist, this could result from two possibilities: either TRAAK adopts two distinct folding patterns during protein synthesis and/or a conformational transition between the two states could occur. The latter would involve a 180-degree rotation of the opposing outer helices, which requires a large energetic penalty to break the existing or form new hydrogen bonds. How such a large-scale conformational

rearrangement could occur after early biosynthesis in the endoplasmic reticulum requires further investigation.

This dynamic cap behaviour may also play a role in the functional diversity of heterodimeric K2P channels. K2Ps can form heterodimers within the same subfamily (*e.g.* TASK-1/TASK-3; TREK-1/TREK-2; THIK-1/THIK-2)[69,70,90,91] and between different subfamilies (e.g. TRESK/TREK-2, TREK-1/TWIK-1; TASK-3/TWIK-1)[92–94] enabling greater functional

**Fig. 4 | S and NS TRAAK cap conformations co-exist in membranes at a ratio influenced by temperature. a** Illustration of the position of hetero spin labelled pair 67R1/144R1 located on one TRAAK subunit in the non-swapped (NS) and swapped (S) conformations (**b**). **c** Raw (background uncorrected) PDS time-domain traces (black) with fit (blue) of heterodimeric TRAAK 67R1/144R1 incubated at 19 °C in the membrane prior to encapsulation with SMALPs, measured over different dipolar evolution times. **d** The best-fit distance distribution (black) was analysed by global fitting in DeerLab from the raw data in **c**. **e** Raw (background uncorrected) PDS time-domain traces (black) with fit (blue) of heterodimeric TRAAK 67R1/144R1 incubated at 40 °C in the membrane prior to encapsulation with SMALPs, measured over different dipolar evolution times. **f** The best-fit distance distribution (black)

was analysed by global fitting in DeerLab from the raw data in (**e**). The distribution of two populations was fitted by a 2-Gaussian fit in DeerLab (red) and respective errors were calculated accordingly, with shaded areas corresponding to the 95% confidence interval (see Methods). The colour bars in d and f represent the reliability of the measured distance ranges (green, shape reliable; yellow, mean and width reliable; orange, mean reliable; red, no quantification possible), the short and long colour bars corresponding to the shortest and longest experimental time window used in **c/e**. An increase of temperature to 40 °C results in a substantially higher population of S state within the TRAAK ensemble, suggesting temperature sensitivity of the cap domain and its potential involvement to TRAAK's heat activation mechanism. Source data are provided as a Source Data file.

diversity, but the functional role of the cap states within those heterodimeric complexes' ensembles is unknown. Given that hetero-dimerisation between different K2P channels could lead to significant alterations in function, it is equally important to monitor whether the NS and S states are present within homo- and hetero- K2P ensembles and to what extent these different physiological conditions could influence such equilibria. Such insights are essential for unravelling the multimodal nature of K2P channels and their roles in disease-related pathways[20,24].

Our findings that the S and NS cap states co-exist in membranes, with their ratio influenced by temperature, suggest that other K2P channels may similarly modulate their cap state to fulfil specific functional roles. It is established that many membrane proteins require specific lipids for modulating their function, folding and maintaining their structural integrity[95]. Detergents are known to strip associated lipids during purification, potentially disrupting these critical interactions. In contrast, our use of a detergent-free SMA-SMALP setup preserves the native lipid environment, enabling co-purification of TRAAK with its associated lipids. This approach allows TRAAK to maintain natural lipid interactions, providing insights into the role of these lipids in its gating mechanism.

Through ES-MS lipid analysis, we identified several unsaturated lipid candidates also present in human plasma membranes of neurons, where TRAAK is natively expressed[17,79] (Fig. 3a, Supplementary Fig. 5). We detected a range of negatively charged lipids, PIs of varying degrees of unsaturation, as well as signalling PIP lipids. We recorded electrophysiology in the presence of these lipids and observed that both lipid types activate TRAAK. Such lipids are found in neuronal membranes and have been reported to affect the function of at least ten distinct K2P channels[68]. This finding is particularly intriguing given that previous quantification of the phospholipid composition in *Pichia pastoris* plasma membranes revealed that PI accounts for only 4.3%, of the total lipid population with no detectable PIP-like lipids[78]. The absence of detectable PIPs in *Pichia* suggests their extreme rarity, falling below the detection threshold in prior studies. While we cannot fully quantify lipid abundances here, our lipidomic data strongly support the fact that the PIPs are highly enriched in our TRAAK-surrounding membrane patches compared to what one would normally see in a whole cell or a plasma membrane fraction. In contrast, DOPE, which has an inverted conical shape and is known to induce membrane curvature, inhibited TRAAK. Finally, POPS did not significantly alter TRAAK's current amplitude, suggesting that this lipid likely plays a structural rather than regulatory role, possibly contributing to the stability of TRAAK within the lipid bilayer.

It is most striking that PC has not been detected in our lipidomics analysis, despite it being the most abundant lipid species in the cells, particularly plasma membranes. Our findings suggest that TRAAK is not associated with PC and has excluded those from its vicinity despite their dominant presence in the native membranes used for expression and in neuronal membranes[78,79], suggesting it forms a very distinct membrane microdomain. Approximately 73% of lipids in the *Pichia pastoris* plasma membrane are unsaturated[78]. The proportion of

unsaturated lipids we detected in the SMALPs encapsulated TRAAK is higher than this percentage and TRAAK seems to be strongly associated with PIPs, which always contain polyunsaturated fatty acids. Our lipidomics data suggest that there is a dominance of the negatively charged headgroup, which is commonly found in species in the inner-leaflet of the plasma membrane. PE and PS lipids favoured by TRAAK are not unusual, and the reason that more saturated fatty acids are found in plasma membrane lipids is to reduce the chance of auto-oxidation of the double bonds.

The ability to detect and quantify the proportional representation of rare states within an ensemble is crucial for understanding K2P channel function. Linking distinct cap states to specific triggers, stimuli, or physiological conditions provides insights into their multimodal behaviour. Our advancement opens the door to unravelling the complex interplay between structure and function in K2P channels and provides a framework to understand their physiological roles and contributions to health and disease.

## Methods
### Cloning and expression
The human *KCNK4* gene (K2P4.1b, residues 1-419, GenBank: AAK49390.1) with mutated surface-exposed cysteine residues (C146S, C206S, C218S, C335S, and C385S) was codon-optimised for expression in *Pichia pastoris* and synthesized by Integrated DNA Technologies. Subsequently, the gene was subcloned into a pPICZ-B expression vector (kindly provided by Dr. Helder Ferreira, University of St Andrews) using the BstBI/AgeI restriction sites. Following that, the gene encoding a His10 tag followed by cysteine-free GFP (cfGFP) (Addgene, plasmid number 54737) and a tobacco etch virus (TEV) with a short linker (SNS) were incorporated upstream of the K2P4.1b (C146S, C206S, C218S, C335S, and C385S) gene. This resulted in the generation of the construct His10-cfGFP-TEV-SNS-K2P4.1b (C146S, C206S, C218S, C335S, and C385S), referred to inCysTRAAK. The wild-type TRAAK was also codon-optimised and generated in an identical vector.

Mutations for the homodimer TRAAK were introduced into the inCysTRAAK construct through site-directed mutagenesis, using primer sequences provided in Supplementary Table 2. The His10 tag sequence of inCysTRAAK was replaced with a V5 tag sequence to facilitate purification of heterodimeric TRAAK mutants. Double mutant pairs—E67C_L144C and E70C_D93C—were then separately introduced into this modified construct, generating two new constructs: V5-cfGFP-TEV-SNS-K2P4.1b_E67C_L144C (C146S, C206S, C218S, C335S, and C385S), referred to as V5-EL, and V5-cfGFP-TEV-SNS-K2P4.1b_E70C_D93C (C146S, C206S, C218S, C335S, and C385S), referred to as V5-ED.

The vectors containing inCysTRAAK and V5-EL (or V5-ED) were linearized and amplified by PCR with the PmeI primers (Supplementary Table 2) and Q5® High-Fidelity DNA Polymerase (New England Biolabs). The linearized plasmids were transformed into *Pichia pastoris* (SMD1163) at a similar ratio by electroporation and plated on a 1 mg/ml zeocin yeast extract peptone dextrose sorbitol (YPDS) plate for 3 days at 30 °C until colonies had grown.

Screening for multi-integrated genes was performed using a combination of both fluorescent and Western blot analysis methods. On the first day, around 50 of the biggest colonies were selected and grown in 0.5 ml YPD media per colony overnight. The next day, each colony was pelleted down and the YPD media discarded before exchange to the induction medium (2× yeast nitrogen base, 0.4 mg/l biotin, 40 mg/l L-histidine, 100 mM potassium phosphate, pH 6.0, 0.5% (v/v) methanol) and further incubated for 8 hours at 28 °C.

After the induction, 200 µl of each colony was inoculated into a 96-well black plate. The fluorescence of each colony was detected under the EVOS fluorescence microscope (Thermofisher). The 6 brightest colonies were selected and normalized to the same amount of cell pellets to approximately 2 g. Those cell pellets were resuspended in around 1 ml PBS, combined with 600 mg of glass beads in 2 ml ground bottom tubes, before beating with a vortex for 3 minutes per cycle. Each colony was beaten for 5 cycles and kept on ice between each cycle for 5 minutes. After lysis, the beads and cell debris were separated by centrifugation at 17,000 × g for 10 minutes. The supernatant of each colony was collected for two Western blots. The primary antibodies were anti-GFP (Antibodies, A283795) and anti-V5 tag (Antibodies, A269661) for each blot, and the secondary antibody was anti-rabbit HRP (Cambridge Bioscience, AC2114) for both blots. ECL detection reagent (Merck Life Science Limited, GERPN2105) was used, and the blots were detected using a Bio-Rad imager. Colonies that showed strong bands in both blots were selected for large-scale expression.

For the large-scale expression of heterodimer TRAAK, the selected colony was inoculated into 10 ml YPD media with 200 µg/ml zeocin and incubated overnight at 30 °C and 200 rpm. Subsequently, the 10 ml overnight culture was transferred to a 2 l baffled flask containing 500 ml of YPD medium and grown for another 8 hours before inoculating into a 4 l minimal medium (2× yeast nitrogen base, 0.4 mg/l biotin, 40 mg/l L-histidine, 100 mM potassium phosphate, pH 6.0) with 1% glycerol. Cells were grown in a shaker at 30 °C and 200 rpm for approximately 24 hours, reaching an OD600 of ~10. The cells were then harvested by centrifugation (3000 × g at 20 °C for 10 min) and resuspended in fresh minimal medium with 0.5% (v/v) methanol to induce protein expression. The cells were further incubated at a lower temperature of 28 °C for another 48 hours. An additional final concentration of 0.5% methanol was added to the minimal culture at 24 hours after induction. Cell pellets were harvested by centrifugation (3000 × g at 4 °C for 30 min), snap-frozen in liquid nitrogen, and stored at −80 °C until further use.

### Isolation of genomic DNA and verification of the integrated gene sequence

The protocol for isolating genomic DNA was adapted from previous reports[96]. Briefly, a 10 ml culture of the selected colony was grown in the presence of 200 µg/ml zeocin overnight. A 200 µl aliquot of the liquid culture was spun down, and the supernatant was discarded. The pellet was resuspended in 100 µl of 200 mM lithium acetate with 1% SDS solution and incubated for 5 minutes at 70 °C. After centrifugation, the supernatant was discarded, and 300 µl of 96–100% ethanol was added to the pellet and vortexed thoroughly. The pellet was spun down at 15,000 × g for 3 minutes and washed with 70% ethanol. After centrifugation to remove the ethanol, the pellet was dissolved in 100 µl of $H_2O$, and the solution was centrifuged to remove cell debris (15 seconds at 15,000 × g). A 2 µl aliquot of the supernatant was used as the template for PCR. The integrated gene fragment containing the V5 tag was amplified from the isolated genomic DNA using the primer pair "V5 tag fragment" listed in Supplementary Table 2. Similarly, the integrated gene fragment containing the His tag was amplified using the primer pair "His-tag fragment" listed in Supplementary Table 2.

### Protein purification and site-directed spin labelling

The frozen cell pellets were disrupted by cryo-milling (Retsch model MM400) in 5 cycles of 3 minutes at 30 Hz and dissolved in the lysis buffer (50 mM Tris, pH 8.0, 500 mM KCl, 0.1 mg/ml soy trypsin inhibitor, 1 mM benzamidine, 2 mM EDTA, 1 mM PMSF) at a ratio of 4 ml of lysis buffer per gram of cell pellet. After cryo-milling, centrifugation at 4696 × g at 4 °C for 50 minutes was performed to remove cell debris and unbroken cells. The resulting supernatant underwent a further centrifugation step at 193,000 × g at 4 °C for 1 hour to obtain membrane pellets. These membrane pellets were resuspended in a solubilisation buffer at a ratio of 1 g pellets per 2 ml solubilisation buffer (composed of 50 mM Tris pH 8.0, 500 mM KCl, protease inhibitor cocktail, 1 mM PMSF, 1 mM benzamidine, 0.1 mg/ml AEBSF). A final concentration of 2.5 mM MTSSL was added to the resuspended membrane pellets, and the mixture was incubated for 1 hour at room temperature. Subsequently, the spin-labelled membrane pellets were subjected to ultracentrifugation at 193,000 × g at 4 °C for 1 hour, followed by a gentle rinse of the pellets with 50 ml of high salt buffer (50 mM Tris, 500 mM KCl). The labelled membrane pellets, resuspended in solubilisation buffer at a concentration of 80 mg/ml, were mixed with an equal volume of solubilisation buffer pre-dissolved with 5% (w/v) SMA co-polymer. This mixture was incubated at room temperature for 1.5 to 2 hours. For encapsulating proteins at 40 °C, the membrane resuspended in solubilisation buffer was placed in a water bath at 40 °C for 30 minutes at a concentration of 80 mg/ml. Following this, it was mixed with a pre-warmed solubilisation buffer containing 5% (w/v) SMA co-polymer and incubated for an additional 30 minutes in the water bath. The non-solubilized material was then separated by ultracentrifugation at 193,000 × g at 4 °C for one hour.

The supernatant post-centrifugation was supplemented with a final concentration of 20 mM imidazole and incubated with 1.5 ml $Ni^{2+}$-NTA beads at 4 °C for 4 hours before loading onto a gravity-flow column. The beads were washed with 50 ml of wash buffer (50 mM Tris pH 8.0, 300 mM KCl, 30 mM imidazole) and eluted with elution buffer (50 mM Tris pH 8.0, 400 mM KCl, 300 mM imidazole). Finally, the elution was subjected to size-exclusion chromatography (SEC) using a Superose6 Increase 10/300 column (Cytiva) equilibrated with SEC buffer (50 mM Tris, pH 8.0, 200 mM KCl). Collected fractions of protein were then concentrated to around 60 µM monomer concentration for subsequent EPR sample preparation. For proteins used for reconstitution, TRAAK in SMALPs was exchanged into the DDM solution. SMALP-encapsulated TRAAK, following elution from SEC, was incubated with 1% DDM in SEC buffer for 30 minutes. A final concentration of 4 mM $MgCl_2$ was then added to the mixture (final buffer conditions: 50 mM Tris pH 8.0, 200 mM KCl, 1% DDM, 4 mM $MgCl_2$), followed by incubation at 4 °C for 2 hours to precipitate the SMA. After the incubation, the aggregated protein and precipitated SMA were pelleted by ultracentrifugation at 100,000 × g.

### Continuous Wave (CW) EPR spectroscopy

The purified spin-labelled sample was loaded into a 3 mm quartz EPR tube for measurement. All cwEPR experiments in solution were measured using a Bruker Magnettech EPR spectrometer ESR5000 operating at X-band and equipped with an integrated temperature controller. The sweep width was set from 330 mT to 345 mT, and the microwave power was set at 10 mW. The modulation frequency was set at 100 kHz, operating at 9.47 GHz frequency, and the modulation amplitude was set to 0.1 mT. The temperature was set to 4 °C to protect the sensitive TRAAK from temperature damage during the multiple accumulation scans that were being performed. 20–50 accumulation scans were collected per sample to acquire high-quality signal-to-noise ratio spectra.

## In silico spin labelling and distance modelling

Pymol-embedded MtsslWizard[97] was used to analyse the labelling sites of TRAAK prior to mutagenesis. All amino acid residues of interest were mutated to cysteine using the Wizard function in PyMOL. MtsslWizard labelled the selected cysteines with the MTSSL spin-labelled side chain R1. A "thorough search" was performed with "VDW restraints" set to "loose" by default. MtsslWizard calculates potential spin label rotamer positions, and an ensemble of 200 rotamer conformations (the maximum number of conformations that can be generated) is presented for each labelling PyMol-embedded site. The distance between two spin labels was calculated by selecting "Measure Mode". In this mode, two ensembles were selected and all possible spin-spin distance vectors between the two spin clusters were calculated. PDB entries 3UM7 (NS) and 4WFF (S) were used to generate the in silico distance distributions for each labelling site under different states.

## PDS distance measurements and data analysis

PDS measurements were performed at an ELEXYS E580 EPR spectrometer from Bruker operating at Q-band (34 GHz) microwave frequency and equipped with a 3 mm cylindrical resonator (QT-II) and a cryogen-free variable temperature cryostat (Cryogenic Ltd, London, United Kingdom). Pulses were amplified by a travelling wave tube amplifier (150 W) (Applied Systems Engineering, United States).

For PDS measurements the 4-pulse $\pi/2(v_A)$-$\tau_1$-$\pi(v_A)$-$(\tau_1 + t)$-$\pi(v_B)$-$(\tau_2 - t)$-$\pi(v_A)$-$\tau_1$-echo sequence was used[98]. Measurements were performed at 50 K with the field position corresponding to the maximum of the field-swept spectrum of the nitroxide. A frequency-offset (pump – detection frequency difference) of +80 MHz and a shot repetition time (SRT) of 2.5 ms were used. $\tau_1$ was set to 380 ns and $\tau_2$ was chosen as transverse dephasing allowed up to 7.5 μs. Pulse lengths were set to 12 ns, 16 ns, and 32 ns for $\pi(v_B)$, $\pi/2(v_A)$ and $\pi(v_A)$, respectively, and optimised for maximum signal. Measurements were performed using an arbitrary waveform generator with a 16-step phase cycle[99]. The pump pulse was set to the maximum of the spectrum and the resonance frequency of the resonator.

PDS data were analysed using DeerAnalysis2022[100] in MATLAB (MathWorks, Natick, MA, United States) using the automated processing under default parameters. To estimate populations of the two distances, a fit of 2 Gaussians was performed in DEERLab[85].

For quantification of the two populations (NS and S state), PDS data of each sample were recorded with different dipolar evolution times and subjected to global fitting of multiple signals from different experiments in DeerLab using Tikhonov regularisation and strong robust generalised cross-validation for the selection of the regularisation parameter. We generally find that two time traces give very good results, but as we had iteratively pushed the dipolar evolution to beyond 7 μs, we had up to 4 traces per sample available and decided to include all of them in the global fit for not introducing bias by deciding which data to discard. Systematic omission of single traces in the global fits of 4 traces leads to no significant changes in the distributions, confirming the self-consistency and robustness of this analysis. A 9 nm distance axis was chosen. Resulting non-parametric distance distributions were further subjected to fitting with a bimodal Gaussian model with peak widths limited to a maximum 1 nm. The relative populations were derived from the weights (amplitudes) in the DeerLab fit, reflecting the peak areas. These weights have uncertainties derived from the model fit in the range of 2-3% representing their 95% (2σ) confidence interval. However, by systematically varying processing parameters (distance axes of 8, 9 or 10 nm), we observed a slightly larger spread in results, amounting to ~1/10th of the weight value of p1 over multiple samples. We thus estimate the true uncertainty to be in this range.

The data analysis for Supplementary Fig. 8-11 was performed using Tikhonov regularisation as implemented in the MATLAB-based DeerAnalysis2022 package. Raw data were loaded into DeerAnalysis2022 and prior to analysis, data points at the end of the time trace were cut off to remove any "2 + 1" end artefacts[56] and data were validated as previously described[41].

## Reconstitution in GUVs for electrophysiology

200 μl of a 10 mg/ml stock of 20% L-α-Phosphatidylcholine (PC) extracted from soybean (Avanti, 541601G–50g) in chloroform was transferred to a glass vial and dried to form a thin film under nitrogen gas. Subsequently, 500 μl of dehydration/rehydration (DR) buffer (5 mM HEPES, pH 7.2 (adjusted with KOH), and 200 mM KCl) was added to the lipid film, followed by bath sonication until the solution became transparent. A final concentration of 0.2% DDM (n-dodecyl-β-D-maltoside) was introduced to the solution and incubated at room temperature for 30 minutes. TRAAK protein was then added to the 2 mg lipids at a protein-to-lipids ratio of 1:100 (wt:wt) and incubated at 4 °C for one hour. Approximately 500 mg of biobeads-SM2 adsorbents, previously washed with water and DR buffer, were added, and the reconstitution was rotated at 4 °C overnight in a 1.5 ml Eppendorf tube. The following morning, the biobeads were separated, and the supernatant was spun down with a Ti55 rotor at 40,000 rpm for 1 hour at 4 °C. The resulting pelleted proteoliposomes were resuspended in 80 μl of DR buffer, snap-frozen in 20 μl aliquots using liquid nitrogen and stored at −80 °C for future use. On the day before the electrophysiology recordings, an aliquot was thawed at room temperature and pipetted onto a glass coverslip inside a petri dish. The petri dish was dried overnight in a desiccator at 4 °C, followed by rehydration with 20 μl of DR buffer for 2 hours at room temperature. Afterward, 2 μl of the rehydrated proteoliposome was added to a 2 ml bath solution (5 mM HEPES, pH 7.2 (adjusted with KOH), 200 mM KCl, and 40 mM MgCl₂) before the recording. A blister was formed immediately and lasted for around 5 hours.

## Patch-clamp electrophysiology recordings

Patch pipettes made of thick-walled borosilicate glass capillaries (World Precision Instruments, 1B150F-6) were pulled to resistances of approximately 3–5 MΩ when filled with a pipette solution (5 mM HEPES, pH 7.2 (adjusted with NaOH), 180 mM NaCl, 20 mM KCl). Recordings were acquired using an Axopatch 200B amplifier (Molecular Devices) with an excised inside-out patch configuration. The currents were filtered at 1 kHz and digitized at 10 kHz using a Digidata 1440 A (Molecular Devices). Negative pressures were applied through a high-speed pressure clamp (ALA Sciences, HSPC-1). Temperature above ambient was controlled using a perfusion temperature controller (Scientifica).

All recordings were conducted in a 10-fold concentration gradient of K⁺ and presented in physiological convention: positive currents indicate K⁺ flux from the high K⁺ concentration intracellular side (bath solution) to the low K⁺ concentration extracellular side.

Arachidonic acid (Sigma; 23401) was aliquoted at 50 mg/ml in ethanol and stored under argon at −20 °C until immediately before use. 16:0-18:1 PS (Avanti; 840034C), 18:0-20:4 PI(4,5)P₂ (Avanti; 850165), and L-α-phosphatidylinositol (Sigma; P6636) were aliquoted at 5 mg/ml in DMSO, and 18:1 PE (Sigma; 76548) was aliquoted at 3 mg/ml in DMSO. All lipids were sonicated until dissolved in DMSO prior to use.

## Electrospray mass spectrometry lipidomic analysis

The lipids were extracted from ~1.1 mg of TRAAK protein that had been purified as described above using SMA and utilised for spin label experiments (One sample was analyzed). This was achieved by 3 successive vigorous extractions with ethanol (90% v/v)[101]. The pooled extracts were dried by nitrogen gas in a glass vial and re-extracted using a modified Bligh and Dyer method[102].

Extracts were dissolved in 15 μl of choloroform:methanol (1:2) and analysed on a Thermo Exploris Orbitrap mass spectrometer by direct infusion in both positive and negative ion modes. Lipid identity based upon accurate mass and MS/MS fragmentation, which was performed using nitrogen as the collision gas with collision energies between 35 and 90 V.

## Atomistic molecular dynamics simulations and energy calculations

The PDB 3UM7 and PDB 4WFF were used as initial models for the NS and S states, respectively. CHARMM-GUI's Membrane Builder tool[103–105] was used to construct the systems with bilayers containing DOPC, DOPE, and POPS lipid molecules. The NS-state membrane protein complex consisted of 332 lipids in total, whereas the S state contained 329 lipids. All simulations employed the CHARMM36 force field[106–108]. The systems were solvated using the TIP3P water model and neutralized with potassium and chloride ions at a concentration of 150 mM. The system composition and dimensions are summarized in Supplementary Table 3. Simulations were performed at two temperatures: 19 °C and 40 °C for each state. Energy minimization was carried out for 5000 steps. Equilibration was conducted in six stages, with positional and dihedral restraints on proteins and lipids gradually released to zero in the final step. Proteins, lipids, and solvent were separately coupled to an external bath using the V-rescale thermostat[109] with a coupling constant of 1.0. Semi-isotropic pressure coupling with a compressibility of $4.5 \times 10^{-5}$ was applied using the stochastic cell-rescale method (Bernetti and Bussi, 2020). Our simulations were performed by combining the cut-off scheme with Force-switch for the Lenard-Jones, where the interactions are smoothly switched between the rvdw switch at 1.0 nm and a cut-off at 1.2 nm. Coulomb interactions were calculated by using the cut-off type PME with a cut-off radius of 1.2 nm. Therefore, the interactions were calculated in real space within the cut-off and in Fourier space using PME after the cutoff. The systems were subjected to production runs of 300–500 ns, using 2 fs time step. All simulations were performed using GROMACS[110] versions 2021.5 and 2024.5 at the CSF3 Manchester and the local Pliotas Lab workstations, accelerated by GPUs. Energy profiles describing non-bonded interactions between the protein and its environment (membrane and solvent) were computed for each state and temperature. Protein, membrane, and solvent were defined as separate energy groups, and the gmx energy tool in GROMACS was used to extract interaction energies. Non-bonded interactions included both Lennard–Jones (LJ) and Coulomb electrostatic (CE) contributions. The approximate interaction energy for each trajectory was estimated by considering the ideal gas as the reference state, as expressed in Eq. 1:

$$\Delta G = G(N, p, T) - G_{\text{ideal gas}}(N, p, T) = -kT \ln(\langle e^{-U/kT} \rangle) \qquad (1)$$

where U is the sum of the LJ and CE interaction energies. PV was calculated by multiplying the reference pressure of 1 bar with the volume, giving a contribution of 93 to 96 kJ/mol for all simulations. This contribution is negligible compared to the energies given in Supplementary Table 4 and was therefore omitted. These calculated interactions represent the effective interaction energy of protein–membrane association. Due to long-time scale fluctuations in the interaction energies, strong energy drifts, a block-averaging scheme was applied in Eq. 1 to determine interaction energy values and associated errors. Differences in energies between simulations of the same state (with identical system composition) but at different temperatures were used to estimate the temperature dependence of the interaction energy change (Supplementary Table 4). Analysis was performed for the last half of the trajectories through the Python scripts using the MDAnalysis package[111,112]. This included the inner leaflet resolved 2D density map for the PS lipids headgroups normalized by the mole fraction of the bulk in the membrane plane, the radial distribution function between the PS lipids in the leaflet and the protein, area per lipid molecule in each leaflet and bilayer thickness. Three independent simulation replicates were performed and analysed, and all resulted in consistent outcomes.

## Reporting summary

Further information on research design is available in the Nature Portfolio Reporting Summary linked to this article.

## Data availability

The lipidomics mass spectrometry raw data generated in this study have been deposited at the Metabolomics Workbench (https://www.metabolomicsworkbench.org/). The EPR and electrophysiology data, as well as the molecular dynamics (MD) analysis outputs generated in this study are provided in the Source Data file. Source data are provided with this paper.

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

## Acknowledgements

This project was supported by a Biotechnology and Biological Sciences Research Council (BBSRC) grant (BB/S018069/1) to C.P. (CW) EPR and PDS measurements were performed at the BioEmPiRe Centre for Structural Biological EPR Spectroscopy, funded by BBSRC (BB/W019795/1) to C.P., and at the St Andrews EPR instrumentation facilities (BBSRC BB/Z516041/1 and BB/T017740/1) and an EPSRC grant (EP/X016455/1) to K.A. and B.E.B.

## Author contributions

Y.M. performed sample preparation and electrophysiology. B.E.B., K.A., and Y.M. performed EPR experiments and analyzed the data. Q.W. and Y.M. carried out MD simulations and analysed the data. T.K.S. performed lipid ES–MS and analyzed the data. V.P. synthesized the SMA copolymer. C.P. obtained funding and conceived the project, and along with Y.M. designed the experiments and wrote the manuscript with input from all authors. C.P. and B.E.B. supervised the study. All authors reviewed and approved the manuscript.

## Competing interests

The authors declare no competing interests.
