## [Peer Review file · Nature Communications]

Swapped and non-swapped TRAAK states co-exist in membranes at a ratio influenced by temperature

Corresponding Author: Dr Christos Pliotas

Version 0:

Reviewer comments:

Reviewer #1

(Remarks to the Author)

In their manuscript entitled “Swapped and non-swapped TRAAK states co-exist in membranes at a ratio influenced by temperature”, the authors demonstrate that these two cap conformations co-exist in the membrane and that their relative proportions are temperature-dependent, with the swapped state being the most prevalent. In addition, the authors present an interesting lipidomic analysis showing that signaling and negatively charged lipids selectively associate with TRAAK, whereas lipids with positively charged head groups, such as phosphatidylcholine, are excluded from the TRAAK membrane microdomain. The topic is of significant interest and will likely attract attention from the community. The authors have supported their experimental findings with structural data to strengthen their claims. However, the manuscript requires revisions before it can be considered for publication.

Major concerns

1. Line 152. While Figure S1b is referenced to support the absence of MTSSL labelling in inCysTRAAK (EPR-silent homodimer), the manuscript does not appear to include a CW-EPR spectrum of the spin-labelled heterodimer (e.g., 67R1/144R1) as a positive control. Including such a spectrum would be valuable to demonstrate successful labelling and signal detection under the same conditions, and to allow a direct visual comparison with the negative control in Figure S1b.
2. While the rainbow colour bars are defined in the figure legends, the manuscript does not discuss how distance-dependent reliability affects the interpretation of the cap state populations. Since the NS state (~3.3 nm) lies within the green “high confidence” zone, and the S state (~5 nm) lies in the less reliable yellow/orange zone, this may influence the precision of population quantification. A brief discussion of this technical limitation would help readers assess the robustness of the S/NS ratio estimation.
3. In several figures (e.g., Fig. 3c, 3e, S6, and S7b), PDS time-domain traces are presented with varying numbers of curves. While Figures 3c and S6 display four traces, Figure 3e shows only two, and Figure S7b shows a single trace. The manuscript does not clearly explain the rationale behind these differences in the number of traces presented. Does the number of curves shown affect the accuracy or confidence in the resulting distance distributions?
4. In Supplementary Figure S7d, the experimental distance distribution derived from the 70R1/93R1 spin pair appears to show only a single peak, despite the in silico models predicting distinct distances for the S and NS states. However, the main text states that two distance components are present (lines 288–289). Please clarify why the manuscript states that two populations are present when only a single peak is observed.
5. A TRAAK concatamer in which Cys67 is introduced in one subunit and Cys144 in the other — that is, placing a single spin label on each subunit at different positions — would serve as a valuable control to distinguish intra-subunit from inter-subunit distance contributions in the PDS measurements.

Minor concerns

1. Lines 88 to 92: Please consider merging both sentences, as the one beginning with “By utilising a unique approach which combines Å-resolution PDS distance measurements...” appears to be grammatically incomplete and lacks a main clause.
2. Line 104: Please include “phosphatidylcholine” before “PC,” as this is the first time the abbreviation is mentioned.
3. Line 115: What do you mean by “species”? Wouldn't “subfamilies” be more appropriate in this context?
4. Line 152: You are citing Figure S1b, not the entire Figure S1.
5. Line 163: Please include “around” before “2 nm,” as the predicted difference between the S and NS states is not exactly 2 nm.
6. Table S1 must include the legend. Ej: What is DM? D-maltoside? Other abbreviations appear in the manuscript but must appear also in the Table to facilitate interpretation.
7. The schematic in Figure 2a labels the spin-labelled subunit as “V5_GFP_TRAAK_L144C”, suggesting it contains only a single cysteine mutation. However, the main text and methods describe the use of a double mutant construct (E67C and L144C) for heterologous single-subunit spin labelling. For clarity and consistency, the figure label should be corrected to reflect both mutations (e.g., “V5_GFP_TRAAK_E67C_L144C” or “V5-EL”).
8. In the legend of Figure 2e, the lipid species “IPC 36:2;O3” is listed, but the meaning of the “;O3” suffix is not explained in the text or figure legend.
9. In Figure 2e, the lipid species “PIP3 42:6-10” is listed, but the meaning of the final “-10” is not explained in the text or figure legend.
10. Line 289: Which part of Figure S7 are you referring to? I assume it is S7d, but this is not clearly stated.
11. In the main text (lines 350–363), you describe the temperature-induced shift in the TRAAK cap state distribution, but you do not cite the corresponding data shown in Figures 3e and 3f. Since these figures directly illustrate the experimental results at 40 °C that support your conclusions, we suggest explicitly citing them in this section to guide the reader to the relevant data
12. Lines 407 to 410 list K2P subfamilies, not families — K2P is the family.
13. From line 364 onward, the text shifts from describing results to interpreting them — discussing the biological relevance of the NS state, potential mechanisms for cap domain transitions, and physiological implications. However, this transition to the discussion section is not explicitly marked. For clarity, especially for readers scanning the manuscript for structure, we suggest including a subheading or transition sentence to clearly delineate the beginning of the discussion.
14. From around line 455 onward, the manuscript revisits several points that were already discussed earlier in the Discussion section — including the energetic considerations for cap transitions, structural flexibility, and the potential regulatory role of the cap domain. These repetitions reduce the impact of the argument and may affect the overall clarity. We suggest revising this section to avoid redundancy and ensure a more concise and focused conclusion.
15. Please check the formatting of “K2P” throughout the document — the “2” should be a subscript.

Reviewer #2

(Remarks to the Author)

Reviewer #3

(Remarks to the Author)

Two-pore domain potassium (K2P) channels generate the leak or background potassium current that sets the resting membrane potential and controls cell excitability. These channels assemble as dimers and feature a unique extracellular cap structure. Structural studies have revealed that this cap can adopt two distinct conformations: a non-swapped (NS) configuration, in which the outer transmembrane helix (TM1) interacts with the inner helix of the same subunit, and a swapped (S) configuration, where TM1 interacts with the inner helix of the adjacent subunit within the dimer.

In this study, the authors used pulsed EPR spectroscopy (PSD) to investigate the conformational dynamics of the human TRAAK channel cap. To do so, they first screened several pairs of spin-labelled cysteine residues and identified the 67R1/144R1 pair as providing the optimal resolution to discriminate between the S and NS states. They further developed a dedicated strategy, termed the HSS-SL protocol, to generate a uniform population of heterodimeric inCysTRAAK/inCysTRAAK channels labelled at these positions.

Prior to the EPR measurements, the authors characterized the lipid composition of TRAAK's immediate membrane environment using electrospray ionization mass spectrometry (ES-MS). They found that TRAAK resides in a lipid microdomain enriched in phosphatidylinositol (PI) and its phosphorylated derivatives (PIP1, PIP2, PIP3), as well as phosphatidylethanolamine (PE) and phosphatidic acid (PA), while phosphatidylcholine (PC) lipids are largely excluded. Using PSD, they demonstrated that both the S and NS cap conformations coexist in TRAAK, with the S-state being predominant. Notably, they showed that the equilibrium between these two states is dynamic and can be modulated by temperature, suggesting that such conformational flexibility could have physiological relevance.

While the study is technically sound and the methodological approach is innovative the physiological relevance is limited. Key experiments are required to strengthen the physiological relevance of the findings.

1. In Fig. 2b, in order to validate the functionality of the constructs used for PSD measurements, it is essential to directly compare the electrophysiological properties of the mutants (including spin-labelled constructs) with the wild-type TRAAK channel. This control is critical to exclude potential artefacts or functional alterations induced by the mutations or spin labelling. Moreover, this control should actually be ideally carried out on the heteromeric tagged and spin-labelled constructs used for PSD.
2. Furthermore, the authors should include a current-voltage (I-V) relationship, ideally obtained using a voltage ramp or step

protocol, to provide a comprehensive characterization of the channel's biophysical properties. This would confirm the functional integrity of the constructs. Additionally, the authors should make sure the regulation of the mutant channels used is unaltered (sensitivity to arachidonic acid, temperature sensitivity mainly).

3. While the PSD results clearly demonstrate that both the S and NS cap conformations coexist in TRAAK, and that their ratio can be modulated by temperature, the direct link between cap swapping and channel activity remains unclear. Specifically, it is not demonstrated that temperature-induced changes in cap conformation is related with an increase in current amplitude.

To further explore the functional relevance of cap swapping, the authors should consider testing gain-of-function mutations known to affect TRAAK gating as well as other stimuli such as arachidonic acid. If such mutations and stimulus alter the S/NS equilibrium, as determined by PSD, and this correlates with changes in channel activity, it would provide strong evidence supporting the physiological relevance of the cap conformational dynamics.

Moreover, to further investigate that NS to S conformational transitions are involved in TRAAK thermal sensitivity, it would be interesting to compare the S to NS ratios of TRAAK at different temperatures over the thermal range to which TRAAK is sensitive (from 30°C to 40°C).

4. Authors carry out an analysis of the nature of lipids surrounding TRAAK. However, they do not clearly expose the aim of this analysis (beyond knowing the lipid partners of TRAAK, which of itself is interesting but constitutes an article on its own) and to what extent it is important within the framework of the article (swapped, non-swapped conformations). Furthermore, authors should provide more in-depth interpretation of their analysis.

Minor points:

- "The equivalent of IPC in human membranes is sphingomyelin, which also has a choline headgroup, and would likely be undetected in our analysis."

Why would it be undetected? The authors mention that PC lipids have previously been detected in similar experiments.

- I.359: "The initial and subsequent enhancement of this dominance of the S state within the TRAAK ensemble upon temperature increase, suggests the S state has greater thermodynamic stability than the NS state." Shouldn't it be the reverse? Upon temperature increase, there is more energy in the system allowing more unstable states to appear?

- Authors speculate that either TRAAK adopts 2 distinct folding patterns during protein synthesis and/or that there is a conformational transition between the two states. But didn't they demonstrate that the latter occurs with their experiment on temperature? (Channels analyzed at low and high temperatures were synthesized with the same protocol.) Authors could mention it.

- I.406: Authors provide an estimate of the swapped to non-swapped conformation transition energy (10 kcal/mol). They should compare it to the energy gain of the system when temperature is increased from 19°C to 40°C in their experiment. Additionally, they could provide a comparison with the energy associated to temperature shifts to which TRAAK is sensitive. This way, the proposition that a mechanism involving cap swapping explains TRAAK thermal sensitivity would be more convincing.

- The paragraph on heterodimers might be over-specified at this point, in particular regarding this sentence: "By quantifying the proportional representation of cap states within these ensembles, we can determine each state's weighted contribution to the channel's functional output."

- In fig2.a, authors should mention the 2 mutations in the vector name: V5_GFP_TRAAK E67C L144C

Mistakes or points of clarification in the text:

- I.156: what do authors mean by "homologous TRAAK sites"?

- I.254: what does the "+ve" mean?

- I.271: PIPS to PIPs

- I.316: contributed to contributing?

Reviewer #4

(Remarks to the Author)

This manuscript probing the swapped (S) and non-swapped (NS) TRAAK potassium ion channel states using DEER spectroscopy is excellent. The research presented clearly indicates that both S and NS states coexist in membranes. The researchers noticed a shift in population to the swapped states at a higher temperature from 19 C to 40 C. The work presented in this study represents a clever way to get a complicated membrane protein sample that has a greatly increased DEER sensitivity for the state of the dimer. The DEER data shown in the paper is excellent. While a Tikhonov regularization is better in general for getting a sense of the most likely distance distribution, (there's no reason to assume that any distribution is purely Gaussian), using a dual gaussian fit is the best way to get a reasonably reliable measurement of the relative populations of the NS and S states (it's what everyone uses for this). I think that this is compelling paper for a Nature Communications publication on the TRAAK K⁺ channel. Also, the methodology could be broadly applied to a lot of biological and membrane systems.

Reviewer #5

(Remarks to the Author)

Potassium channels are oligomers of transmembrane subunits. The membrane-spanning domain of a multimeric channel is an assembly of alpha helices that interact with each other and with membrane bilayer lipids. The pore selectivity filter is formed by four reentrant domains which themselves interact with parts of the transmembrane helices. This assembly is

highly stable - an essential condition for pore geometry and channel selectivity. Studies to date show that the oligomerization process takes place very early in channel biosynthesis in the endoplasmic reticulum, under the control of chaperone proteins. No studies have shown that the composition or overall conformational organization of the mature complex can be altered at the plasma membrane, probably because this would require too much energy. Under physiological conditions, very minor conformational changes modify the activity or selectivity of the channel and regulate its function.

Two-pore domain potassium (K2P) channels, including TRAAK, are dimers of subunits. Previous structural studies have shown that mature K2P channels can exist in two conformational states, the so-called non-swapped (NS) and swapped (S) forms. The two forms differ in the positions of the helices forming the extracellular cap and the membrane-spanning domains of each subunit in the dimer. The proportion of these two forms in cells and their functional significance are not known.

In this manuscript, Ma and colleagues expressed TRAAK in yeast, purified them and used pulse dipolar electron paramagnetic resonance spectrometry to calculate the proportion of N and NS forms. They also show that temperature influences this ratio suggesting that the previously reported effect of temperature on TRAAK could be due to a conversion between the N and NS forms. I'm not familiar with the technique used and the possible artifacts that it may produce, but the conclusions that TRAAK can switch between the N and NS states and that this switch has physiological significance are not supported by the provided data at this stage. Switching between the S and NS forms would require major conformational changes, with the unfolding and refolding of several domains and the modification of a considerable numbers of hydrophilic and hydrophobic interactions. No kinetics of this process are provided in the manuscript. If such rearrangement is possible after biosynthesis, then it is a multi-step process that should be slow. The effect of temperature on TRAAK/TREK1/TREK2 channel activity is rapid, as the effects of other stimuli (stretch, lipids, pH...). Also a 10-fold increase in TRAAK current is measured for a change from 17 to 40°C, while an increase of only 15% in the S form is observed. Nor is there any functional evidence that N and NS differ in their open probability or conductance, which would support the hypothesis of the authors.

In this work, the authors also took advantage of the way TRAAK is purified by SMA encapsulation to analyze the lipids surrounding the channel. While this is a promising approach, it has no bearing on the main message of the manuscript and its title. Much more work will be needed to show whether what is observed in yeast is relevant in mammalian cells for TRAAK function and/or whether other K2Ps behave differently or similarly to TRAAK with regard to the recruitment of surrounding lipids.

In conclusion, the authors have used novel and elegant approaches to study the structure of TRAAK and its lipid neighborhood. But this work is still too preliminary to draw any firm conclusions about a functional role of the N and NS states of K2P channels.

Reviewer #6

(Remarks to the Author)

Version 1:

Reviewer comments:

Reviewer #1

(Remarks to the Author)

I do not have any further comments for the authors. All my concerns have been addressed.

Reviewer #2

(Remarks to the Author)

Reviewer #3

(Remarks to the Author)

In their revised manuscript and their answers to the points we raised, Ma et al. have provided extensive supplementary data, additional details and justifications that strengthen their study on the temperature-dependent existence of the swapped and non-swapped conformations of the K2P TRAAK channel.

More specifically, authors successfully demonstrated through patch-clamp recordings that the mutated version of TRAAK used for their pulse dipolar EPR spectroscopy experiments is functional and retains the main regulatory properties of the wild type channel. We are now convinced of the scientific soundness of their experimental approach.

Regarding the part where authors study the lipids surrounding TRAAK, the authors clarified their analysis and answered our

questions satisfactorily.

Finally, the authors successfully addressed our concerns regarding energetic considerations. The details they added in the manuscript as well as their molecular dynamics study largely improve the understanding of these energetic aspects.

However, even if authors clearly explain why it is technically challenging to assess the physiological relevance of the coexistence of the two TRAAK conformations (swapped and non-swapped), they failed to provide convincing evidence for the physiological significance of their findings.

Overall, their results represent a clear breakthrough in the field of K2P channel structural biology, and their approach is technically innovative. However, the limited physiological relevance of their findings may constrain the potential readership and overall impact of the study.

Reviewer #5

(Remarks to the Author)

We thank the authors for their response. As they agreed that "domain swapping at the plasma membrane would require breaking strong inter-helical bonds which would be energetically unfavourable and likely destabilizing to the pore domain" and that it is therefore reasonable to "question the likelihood of a large-scale conformational rearrangement after early biosynthesis in the ER", it is important to mention this point in the manuscript as something to keep in mind for future studies.

Reviewer #6

(Remarks to the Author)

Reviewer #7

(Remarks to the Author)

This manuscript shows that swapped (S) and non-swapped (NS) TRAAK states co-exist in membranes and that their population shifts with temperature, supported by careful PDS/DEER measurements. The MD simulations aim to rationalize the temperature dependence by estimating "free energies" from non-bonded interaction energies. However, the current MD analysis has methodological and interpretational issues that require major revision.

Major Comments:

The statement that Coulombic interactions were "shifted to zero between 0–12 Å" implies that long-range electrostatics were truncated rather than treated with Ewald summation. For a membrane-protein system, this is not acceptable because electrostatics strongly govern protein–lipid and protein–solvent interactions. All simulations must be rerun with Particle Mesh Ewald (PME) to appropriately treat long-range interactions.

The "approximate free energy" defined from total non-bonded interaction energy using an ideal-gas reference (Eq. 1) is unconventional and thermodynamically inconsistent. Entropic and PV terms are neglected, yet the quantity is interpreted as a free energy. Either rename this metric as an interaction-energy descriptor and avoid free-energy language, or replace it with a recognized method (alchemical FEP/TI, or a PMF along a physically motivated coordinate). If PV is omitted, provide quantitative justification.

The molecular origin of the temperature-dependent changes in non-bonded interactions remains unclear. Do these shifts primarily reflect membrane mechanics (e.g., changes in bilayer thickness, area per lipid, or leaflet asymmetry), or differences in the stability of the membrane-exposed cap? What is the pressure profile along the membrane normal? Please quantify and discuss how these properties vary with temperature for both S and NS.

The simulated bilayers contain DOPC:DOPE:POPS (72:14:14), i.e., an explicit fraction of phosphatidylserine (PS). Because PS is negatively charged and typically enriched in the inner leaflet, its spatial organization relative to TRAAK is crucial for interpreting the reported protein–membrane interaction energies. The manuscript should show where PS resides relative to the protein and whether it engages specific basic patches on the channel surface. Please provide leaflet-resolved 2D density maps of PS headgroups in the membrane plane, normalized by the bulk PS mole fraction, for both S and NS (with an inset radial profile $g(r)$ to quantify enrichment). Comparing these metrics between S and NS would clarify whether the reported energetic differences reflect PS-mediated contacts or generic bilayer mechanics.

Minor Comments:

TRAAK gating is tension-sensitive. make the effective tension depend on leaflet lipid numbers and box-area fluctuations. Please clarify inner/outer leaflet lipid counts.

A representative snapshot showing the protein within the simulation box and the bilayer dimensions would help readers understand boundary conditions and system packing.

Version 2:

Reviewer comments:

Reviewer #7

(Remarks to the Author)

I do not have any further comments for the authors. All my concerns have been addressed.

Response to the comments from the Reviewers

We sincerely appreciate the insightful and constructive comments from the Reviewers, which have greatly helped us improve our manuscript. Below is our point-by-point response to the comments. The comments from the Reviewers are shown in blue, our responses are in black, and the changes made in the revised manuscript are highlighted in yellow.

Reviewer #1 (Remarks to the Author):

In their manuscript entitled “Swapped and non-swapped TRAAK states co-exist in membranes at a ratio influenced by temperature”, the authors demonstrate that these two cap conformations co-exist in the membrane and that their relative proportions are temperature-dependent, with the swapped state being the most prevalent. In addition, the authors present an interesting lipidomic analysis showing that signaling and negatively charged lipids selectively associate with TRAAK, whereas lipids with positively charged head groups, such as phosphatidylcholine, are excluded from the TRAAK membrane microdomain. The topic is of significant interest and will likely attract attention from the community. The authors have supported their experimental findings with structural data to strengthen their claims. However, the manuscript requires revisions before it can be considered for publication.

Response:

We sincerely appreciate the positive comments made by Reviewer #1 and we are pleased that the Reviewer believes that the community will find our study of significant interest and that it will attract considerable attention.

Major concerns

1. Line 152. While Figure S1b is referenced to support the absence of MTSSL labelling in inCysTRAAK (EPR-silent homodimer), the manuscript does not appear to include a CW-EPR spectrum of the spin-labelled heterodimer (e.g., 67R1/144R1) as a positive control. Including such a spectrum would be valuable to demonstrate successful labelling and signal detection under the same conditions, and to allow a direct visual comparison with the negative control in Figure S1b.

Response:

We agree with Reviewer 1 that the spectra of the spin-labelled heterodimeric mutants should be also presented alongside the other spin-labelled homodimers. We have now included this data in the new **Supplementary Figure 6** in the revised manuscript (reproduced as Figure 1 below). The data further confirm that these hetero-sites can be spin-labelled and also allow for direct comparison between homo- and heteromeric labelled TRAAK mutants (i.e., 67R1, 144R1 and 67R1/144R1).

Figure 1: CW-EPR spectra of spin-labelled TRAAK homodimers and heterodimers in SMALPs. Continuous-wave (CW) EPR spectra recorded at 4 °C for various spin-labelled TRAAK constructs incorporated into SMALPs. Spectra correspond to the labelling sites indicated in the figure. R1 denotes the nitroxide side chain formed by the reaction of MTSSL with engineered cysteine residues.

2. While the rainbow colour bars are defined in the figure legends, the manuscript does not discuss how distance-dependent reliability affects the interpretation of the cap state populations. Since the NS state (~3.3 nm) lies within the green “high confidence” zone, and the S state (~5 nm) lies in the less reliable yellow/orange zone, this may influence the precision of population quantification. A brief discussion of this technical limitation would help readers assess the robustness of the S/NS ratio estimation.

Response:

The Reviewer is correct to mention the technical limitation of the quantification of the cap states. We have now added a brief discussion explaining the limits of our method. This will help the reader to follow the rationale of fitting the population with a bimodal Gaussian model.

Change 1 on “Results”:

NS and S states co-exist in membranes and their ratio is influenced by temperature

The experimental PDS distance distributions are in excellent agreement with predictions for the NS and S cap states (Fig. 4, Fig. S7, S8). For quantification of populations the shape of the distance distribution must be reliable (highest reliability range, green colour bars in Fig. 4d, f)⁸². To increase the reliability of the distance distributions we combined short time traces with high time resolution and excellent signal-to-noise with longer time traces and analysed them globally^{85,86}. While usually two traces are sufficient, we decided to include all data available (up to 4 traces) to avoid bias by selecting certain subsets of data. Despite multiple different analysis workflows and controls consistently reproducing these two peaks in the non-parametric distributions, the peak representing the S state lies outwith the range where the shape of the distribution can be directly interpreted (this being ultimately limited by PDS trace length and thus echo lifetime). To overcome this technical limitation, we fitted these non-parametric distance distributions with a bimodal Gaussian model. While the quantification is having inherently higher uncertainty for longer distances, we found highly consistent results using technical and biological replicates. Estimating the relative populations of NS and S cap states we found that the majority of TRAAK is in the S state, with a substantial minority population being in the NS state S/NS: (57±4 and 43±4)% (with the uncertainty representing 95% confidence or 2σ). The dominance of the S state within the TRAAK ensemble aligns with the overwhelmingly frequent occurrence in the previous x-ray crystallography and cryoEM structures of TRAAK and other K2P channels. Our findings are also consistent with previous computational analysis suggesting the S state is energetically favourable⁸⁷, thereby more stable and amenable to structural studies.

Change 1 on “Methods”:

For quantification of the two populations (NS and S state), PDS data of each sample were recorded with different dipolar evolution times and subjected to global fitting of multiple signals from different experiments in DeerLab using Tikhonov regularisation and strong robust generalised cross-validation for the selection of the regularisation parameter. We generally find that two time traces give very good results but as we had iteratively pushed the dipolar evolution to beyond 7 μs we had up to 4 traces per sample available and decided to include all of them into the global fit for not introducing bias by deciding which data to discard. Systematic omission of single traces in the global fits of 4 traces leads to no significant changes in the distributions confirming the self-consistency and robustness of this analysis. A 9 nm distance axis was chosen. Resulting non-parametric distance distributions were further subjected to fitting with a bimodal Gaussian model with peak widths limited to maximum 1 nm. The relative populations were derived from the weights (amplitudes) in the DeerLab fit reflecting the peak areas. These weights have uncertainties derived from the model fit in the range of 2-3% representing their 95% (2σ) confidence interval. However, by systematically varying processing parameters (distance axes of 8, 9 or 10 nm), we observed a slightly larger spread in results amounting to ~1/10th of the weight value of p1 over multiple samples. We thus, estimate the true uncertainty to be in this range.

3. In several figures (e.g., Fig. 3c, 3e, S6, and S7b), PDS time-domain traces are presented with varying numbers of curves. While Figures 3c and S6 display four traces, Figure 3e shows only two, and Figure S7b shows a single trace. The manuscript does not clearly explain the rationale behind these differences in the number of traces presented. Does the number of curves shown affect the accuracy or confidence in the resulting distance distributions?

Response:

Ideally, what one would want is measuring a single PDS trace at sufficient length for resolving all populations (highest reliability range, green colour bar) and excellent signal-to-noise ratio for processing with high confidence. A shorter trace with small time steps can resolve the distance distribution for short distances whereas long times are required for reliable background extraction and reliable long-distance populations. Measuring a long trace with short time steps and excellent signal-to-noise is experimentally not always feasible (with the exception of the sample shown in Figure S8 in the revised manuscript where it was not possible to resolve the two populations even though both are in the highest reliability range of the distribution) and this has been demonstrated and discussed in literature (e.g., Rein *et al. J. Magn. Reson.* **2018**, 295, 17-26 and Fábregas Ibáñez *et al. Magn. Reson.* **2020**, 1, 209-224). We have measured multiple traces with increasing time windows of most samples to increase confidence that both cap states are indeed present.

Thus, our approach to this limitation was the use of global fitting of the data in DeerLab. The actual number of traces is dependent on the data we had available – for the repeat of the 67R1/144R1 mutant (shown in Fig S7 in revised manuscript) we stayed close to the original sample (Fig 4c in revised manuscript). For the heated sample (Fig 4e in revised manuscript) it was more challenging to measure at longer dipolar evolution times, thus we settled with two traces. Generally, we find that two traces suffice but as we had more data, we did not want to bias the analysis by selecting which data to exclude from the analysis.

Analysing data for the samples with the 4 traces, leaving out one trace at a time, yielded essentially the same distributions.

We have added clarifying statements to the Results section of the manuscript (**Change 1 on “Results”**) and the Methods section of the SI (**Change 1 on “Methods”**).

4. In Supplementary Figure S7d (Fig S8d in revised manuscript), the experimental distance distribution derived from the 70R1/93R1 spin pair appears to show only a single peak, despite the *in silico* models predicting distinct distances for the S and NS states. However, the main text states that two distance components are present (lines 288–289). Please clarify why the manuscript states that two populations are present when only a single peak is observed.

Response:

We thank the Reviewer for pointing this out. We have now revised the wording in the main text accordingly:

Change 2 on “Results”:

PDS combined with HSS-SL enables probing of the entire TRAAK cap state ensemble

Prior to PDS, continuous-wave EPR (cw-EPR) confirmed that the mutated sites in each expressed protein were efficiently spin-labelled and that any excess labelling reagent was quantitatively removed (Fig. S6). We then conducted Q-band (34 GHz) PDS distance measurements on the 67R1/144R1 pair (Fig. 4a, b) captured in native cell lipid membranes (SMALPs) at near ambient temperature (19°C). The time traces exhibited strong visual oscillations, which led to robust distance distributions of two populations centred at 3.3 nm and 5 nm respectively with different weights (Fig. 4c, d). Widths and shapes of these distance populations are in full agreement with the *in silico* predictions for the cap NS and S states, suggesting both states exist within the conformational ensemble of TRAAK in the lipid membrane. These measurements were reproduced for a different protein batch yielding almost identical PDS results (Fig. S7). To further validate our findings are not dependent upon a single spin pair combination⁸², we performed PDS on a second heterologously spin labelled pair (70R1/93R1). For this pair we also observed strong oscillations in the raw traces, indicating reliability of the resulting distance distributions (Fig. S8). However, the separation between S and NS modelled distances for this pair is not as ideal as for 67R1/144R1 (Fig. S8d), due to limitations by the unique structural architecture of the K2P cap. Modelled distance distributions show significant overlap, thus the experimentally obtained trace and corresponding distance distribution cannot resolve the presence of two distinct distance components. Nevertheless, the width of the observed distribution almost perfectly matches the combined width of the two modelled distributions, while the presence of only one of the states should have yielded a substantially narrower distribution with a shift in distance for the maximum probability. These data further support the presence of both S and NS species in the TRAAK ensemble.

5. A TRAAK concatamer in which Cys67 is introduced in one subunit and Cys144 in the other — that is, placing a single spin label on each subunit at different positions — would serve as a valuable control to distinguish intra-subunit from inter-subunit distance contributions in the PDS measurements.

Response:

We appreciate this suggestion, and it is indeed a valuable point. However, constructing a new concatamer is not straightforward, as it requires optimisation of the linker between the two subunits. An inappropriate linker may introduce artifacts that could alter the native conformation of TRAAK and is out of the scope of this study. Nonetheless, we will consider implementing this approach in a future study.

Minor concerns

1. Lines 88 to 92: Please consider merging both sentences, as the one beginning with “By utilising a unique approach which combines Å-resolution PDS distance measurements...” appears to be grammatically incomplete and lacks a main clause.

Change 3 on “Results”:

Pulse Dipolar electron paramagnetic resonance Spectroscopy (PDS) is a powerful technique to monitor dynamics, folding, oligomerisation and conformational ensembles in integral membrane proteins⁴¹⁻⁵⁵. This ensemble method commonly involves the incorporation of paramagnetic spin labels through cysteine-based modifications on selected protein sites to determine inter-spin distance distributions⁵⁶⁻⁵⁹. We utilised a unique approach, which combines Å resolution PDS distance measurements in ion channels^{41,42,55,60,61} with heterologous single-subunit spin labelling (HSS-SL). This strategy, we purposely developed here, was essential to distinguish between the two cap states of TRAAK (Fig. 1b,c). We unbiasedly monitored the complete conformational ensemble of the TRAAK cap domain. We studied human TRAAK, for which both the S and NS

2. Line 104: Please include “phosphatidylcholine” before “PC,” as this is the first time the abbreviation is mentioned.

Corrected

3. Line 115: What do you mean by “species”? Wouldn’t “subfamilies” be more appropriate in this context?

Corrected

4. Line 152: You are citing Figure S1b, not the entire Figure S1.

Corrected

5. Line 163: Please include “around” before “2 nm,” as the predicted difference between the S and NS states is not exactly 2 nm.

Corrected

6. Table S1 must include the legend. Ej: What is DM? D-maltoside? Other abbreviations appear in the manuscript but must appear also in the Table to facilitate interpretation.

Corrected

7. The schematic in Figure 2a labels the spin-labelled subunit as “V5_GFP_TRAAK_L144C”, suggesting it contains only a single cysteine mutation. However, the main text and methods describe the use of a double mutant construct (E67C and L144C) for heterologous single-subunit spin labelling. For clarity and consistency, the figure label should be corrected to reflect both mutations (e.g., “V5_GFP_TRAAK_E67C_L144C” or “V5-EL”).

Corrected

8. In the legend of Figure 2e, the lipid species "IPC 36:2;O3" is listed, but the meaning of the ";O3" suffix is not explained in the text or figure legend.

Corrected

9. In Figure 2e, the lipid species "PIP3 42:6-10" is listed, but the meaning of the final "-10" is not explained in the text or figure legend.

Corrected

10. Line 289: Which part of Figure S7 are you referring to? I assume it is S7d, but this is not clearly stated.

Corrected

11. In the main text (lines 350–363), you describe the temperature-induced shift in the TRAAK cap state distribution, but you do not cite the corresponding data shown in Figures 3e and 3f. Since these figures directly illustrate the experimental results at 40 °C that support your conclusions, we suggest explicitly citing them in this section to guide the reader to the relevant data

Corrected

12. Lines 407 to 410 list K2P subfamilies, not families — K2P is the family.

Corrected

13. From line 364 onward, the text shifts from describing results to interpreting them — discussing the biological relevance of the NS state, potential mechanisms for cap domain transitions, and physiological implications. However, this transition to the discussion section is not explicitly marked. For clarity, especially for readers scanning the manuscript for structure, we suggest including a subheading or transition sentence to clearly delineate the beginning of the discussion.

Corrected, we added “Discussion” as a subheading.

14. From around line 455 onward, the manuscript revisits several points that were already discussed earlier in the Discussion section — including the energetic considerations for cap transitions, structural flexibility, and the potential regulatory role of the cap domain. These repetitions reduce the impact of the argument and may affect the overall clarity. We suggest revising this section to avoid redundancy and ensure a more concise and focused conclusion.

We thank the Reviewer for their suggestions. In response, we have removed the section on energetic considerations (now replaced by our more robust MD energy calculations), and eliminated repetitive text to ensure our main messages remain focused and undiluted.

15. Please check the formatting of “K₂P” throughout the document — the “2” should be a subscript.

Both K₂P and K₂P abbreviations are widely accepted in literature, and we use “K₂P” throughout our manuscript.

Reviewer 2:

This study characterizes the populations of the swapped and nonswapped conformational states of the TRAAK channel cap in a near native lipid environment, using cysteine-based spin labeling. Additionally, the authors examine the temperature dependence of these conformational states and carry out a lipidomic analysis of lipids in close proximity to TRAAK channels. The research addresses multiple relevant aspects, and it is laudable that the authors present a comprehensive and integrative study, combining both experimental and theoretical approaches. There are, however, several points that could be further clarified or expanded upon to strengthen the study:

Major concerns

1. Lines 295, 296: It is not clear how it is claimed that the SMA used forms 10 nm lipid disks. While Postis et al. (2015) cite a maximal diameter of ~15 nm and Lee et al. (2016) provide no size data, both ultimately reference Jamshad et al. (2014) (not directly quoted in your text), which reports diameters: ~9.8 and 11–16 nm by different techniques. Please mention the source of the value mentioned (e.g., your own measurements, Jamshad data or estimates). This point is key, because it will provide further evidence for your interpretations of the PDS distances for intermolecular interactions.

Response:

We thank the Reviewer for their point and agree that the size of the SMA lipid discs is inherently variable. Previous studies highlight how the SMA copolymer effectively solubilizes membranes and can create discoidal assemblies of varying sizes, with measurements typically ranging from 10 nm to 16 nm in diameter, depending on the lipid environment and the presence of proteins (Long et al., 2013; Jamshad et al., 2015). This variability underscores the adaptability of the SMA discs to different lipid compositions.

The SMA-based nanodiscs can support various native lipid-protein interactions, thereby preserving their functional state and activity. The adaptability of SMA disks can thus be leveraged to optimize conditions for studying such protein-lipid interactions, ensuring the physiological relevance of these environments is maintained (Dörr et al., 2014). Moreover, even for the larger SMA diameter reported of 16 nm two TRAAK channels cannot be accommodated in a single SMA disc (i.e., TRAAK has a diameter of ~8 nm, *Brohawn et al., Science, 2012*), thus diminishing any unspecific inter-dimer distance contributions from within the same membrane disc, which could obscure our PDS experiments. In our study, we further

ensured that two TRAAK channels cannot be inserted in a single SMA disc as we selected only the TRAAK fractions consistent with the purified monodisperse peak of a single dimeric TRAAK disc size following size exclusion chromatography, and before subjecting to subsequent EPR experiments (i.e., CW and PDS).

We now made a change in the revised manuscript to reflect these points.

Change 4 on “Results”:

We used a SMA type which forms lipid discs ranging from 10 to 16 nm in diameter, depending upon the lipid environment and the presence of proteins^{83,84}. Such adaptability of SMA is crucial for maintaining the functional activity of TRAAK in the membrane. As the TRAAK's cross sectional TM region being approximately 8 nm, means only one TRAAK channel could fit in each lipid disc. We performed PDS for all four homo-dimeric single Cys TRAAK mutants for all the individual sites included in our heterodimeric pairs (i.e. 67R1, 70R1, 93R1, L144R1) to demonstrate efficient formation of TRAAK homodimers and rule out intermolecular distance contributions. We subsequently modified different variants of homodimeric TRAAK with MTSSL and performed PDS distance measurements in SMA (Fig. S9, S10). For all single Cys homodimeric TRAAK mutants tested we observed strong dipolar coupling interactions between the singly labelled homo-dimeric channels and obtained reliable distance distributions. Distances are highly consistent with the *in silico* predictions for the two species which are expected to overlap, though with different mean values for the different sites and with no additional distance

References:

Dörr, J., Koorengel, M., Schäfer, M., Prokofyev, A., Scheidelaar, S., Crujisen, E., ... & Killian, J. (2014). Detergent-free isolation, characterization, and functional reconstitution of a tetrameric k⁺ channel: the power of native nanodiscs. *Proceedings of the National Academy of Sciences*, 111(52), 18607-18612. <https://doi.org/10.1073/pnas.1416205112> [doi.org]

Jamshad M, Grimard V, Idini I, Knowles TJ, Dowle MR, Schofield N, Sridhar P, Lin YP, Finka R, Wheatley M, Thomas OR, Palmer RE, Overduin M, Govaerts C, Ruyschaert JM, Edler KJ, Dafforn TR. Structural analysis of a nanoparticle containing a lipid bilayer used for detergent-free extraction of membrane proteins. *Nano Res.* 2015 Mar;8(3):774-789. doi: 10.1007/s12274-014-0560-6. Epub 2014 Oct 23. PMID: 31031888; PMCID: PMC6485620.

Long, A., O'Brien, C., Malhotra, K., Schwall, C., Albert, A., Watts, A., ... & Alder, N. (2013). A detergent-free strategy for the reconstitution of active enzyme complexes from native biological membranes into nanoscale discs. *BMC Biotechnology*, 13(1). <https://doi.org/10.1186/1472-6750-13-41> [doi.org]

2. It is recommended to include the crystallographic structures used in the *in silico* analysis (e.g. TRAAK 3UM7 for the NS state and 4WFF for S state) to allow replicability. Also, it would be valuable to specify the number of conformations generated (n, sample size) to construct the distance distributions, and to report descriptive statistics (mean and standard deviation) directly in the text. Also, the distributions presented have a perfect Gaussian curve, perhaps you used some additional adjustment that should be mentioned. This would help to reinforce the transparency and statistical robustness of the *in-silico* results presented.

Response:

We appreciate the suggestions and have now added a descriptive sentence in the Methods section to indicate the crystallographic structures used for the *in silico* analysis and the number of conformations generated. Mean distances of our *in silico* models are also included in Figure 1b and c. Both the mean distance and the SD are provided in the main text. Regarding the details of obtaining the bimodal Gaussian model, we have added clarifying statements to the Methods section of the SI.

Change 1 on “Methods”:

For quantification of the two populations (NS and S state), PDS data of each sample were recorded with different dipolar evolution times and subjected to global fitting of multiple signals from different experiments in DeerLab using Tikhonov regularisation and strong robust generalised cross-validation for the selection of the regularisation parameter. We generally find that two time traces give very good results but as we had iteratively pushed the dipolar evolution to beyond 7 μ s we had up to 4 traces per sample available and decided to include all of them into the global fit for not introducing bias by deciding which data to discard. Systematic omission of single traces in the global fits of 4 traces leads to no significant changes in the distributions confirming the self-consistency and robustness of this analysis. A 9 nm distance axis was chosen. Resulting non-parametric distance distributions were further subjected to fitting with a bimodal Gaussian model with peak widths limited to maximum 1 nm. The relative populations were derived from the weights (amplitudes) in the DeerLab fit reflecting the peak areas. These weights have uncertainties derived from the model fit in the range of 2-3% representing their 95% (2σ) confidence interval. However, by systematically varying processing parameters (distance axes of 8, 9 or 10 nm), we observed a slightly larger spread in results amounting to $\sim 1/10^{\text{th}}$ of the weight value of p1 over multiple samples. We thus, estimate the true uncertainty to be in this range.

Change 2 on “Methods”:

In silico spin labelling and distance modelling

PyMol-embedded MtsslWizard⁹⁷ was used to analyse the labelling sites of TRAAK prior to mutagenesis. All amino acid residues of interest were mutated to cysteine using the Wizard function in PyMoL. MtsslWizard labelled the selected cysteines with the MTSSL spin labelled side chain R1. A "thorough search" was performed with "VDW restraints" set to "loose" by default. MtsslWizard calculates potential spin label rotamer positions, and an ensemble of 200 rotamer conformations (the maximum number of conformations that can be generated) is presented for each labelling PyMol-embedded site. The distance between two spin labels was calculated by selecting "Measure Mode". In this mode, two ensembles were selected and all possible spin-spin distance vectors between the two spin clusters were calculated. PDB entries 3UM7 (NS) and 4WFF (S) were used to generate the *in silico* distance distributions for each labelling site under different states.

Change 5 on “Results”:

the highest resolution (Fig. 1c, Fig. S4). The predicted difference in the most probable distances between S and NS states is around 2 nm (mean \approx 5 nm, SD \approx 0.3 nm for S state; mean \approx 3.3 nm, SD \approx 0.5 nm for NS state) allowing unambiguous separation and quantification of the two states⁷⁵. (Fig. 1). This separation is significantly greater than twice the length of the R1 side-chain, thus beyond and excluding any biased spin label conformation that could obscure our PDS measurements. The predicted distances also fall within the optimal distance range for PDS, which is between 2 to 6 nm, allowing the two states to be distinguished.

Reviewer 3:

Two-pore domain potassium (K2P) channels generate the leak or background potassium current that sets the resting membrane potential and controls cell excitability. These channels assemble as dimers and feature a unique extracellular cap structure. Structural studies have revealed that this cap can adopt two distinct conformations: a non-swapped (NS) configuration, in which the outer transmembrane helix (TM1) interacts with the inner helix of the same subunit, and a swapped (S) configuration, where TM1 interacts with the inner helix of the adjacent subunit within the dimer.

In this study, the authors used pulsed EPR spectroscopy (PDS) to investigate the conformational dynamics of the human TRAAK channel cap. To do so, they first screened several pairs of spin-labelled cysteine residues and identified the 67R1/144R1 pair as providing the optimal resolution to discriminate between the S and NS states. They further developed a dedicated strategy, termed the HSS-SL protocol, to generate a uniform population of heterodimeric inCysTRAAK/inCysTRAAK channels labelled at these positions.

Prior to the EPR measurements, the authors characterized the lipid composition of TRAAK's immediate membrane environment using electrospray ionization mass spectrometry (ES-MS). They found that TRAAK resides in a lipid microdomain enriched in phosphatidylinositol (PI) and its phosphorylated derivatives (PIP1, PIP2, PIP3), as well as phosphatidylethanolamine (PE) and phosphatidic acid (PA), while phosphatidylcholine (PC) lipids are largely excluded.

Using PDS, they demonstrated that both the S and NS cap conformations coexist in TRAAK, with the S-state being predominant. Notably, they showed that the equilibrium between these two states is dynamic and can be modulated by temperature, suggesting that such conformational flexibility could have physiological relevance.

While the study is technically sound and the methodological approach is innovative the physiological relevance is limited. Key experiments are required to strengthen the physiological relevance of the findings.

Response:

We sincerely appreciate Reviewer 3 recognising the innovative aspects of our method. We have now performed additional electrophysiology experiments and include in the revised manuscript to better understand channel function and contribute to knowledge towards the physiological relevance of our PDS findings. These new data are consistent with our initial conclusions and shed light into the role of native membrane lipids (i.e., identified by our ES-MS analysis) into TRAAK's function. We do hope that these new functional observations will satisfy the Reviewer and we believe these further strengthen our study.

1. In Fig. 2b, in order to validate the functionality of the constructs used for PDS measurements, it is essential to directly compare the electrophysiological properties of the mutants (including

spin-labelled constructs) with the wild-type TRAAK channel. This control is critical to exclude potential artefacts or functional alterations induced by the mutations or spin labelling. Moreover, this control should actually be ideally carried out on the heteromeric tagged and spin-labelled constructs used for PDS.

2. Furthermore, the authors should include a current-voltage (I-V) relationship, ideally obtained using a voltage ramp or step protocol, to provide a comprehensive characterization of the channel's biophysical properties. This would confirm the functional integrity of the constructs. Additionally, the authors should make sure the regulation of the mutant channels used is unaltered (sensitivity to arachidonic acid, temperature sensitivity mainly).

Response to 1 &2:

We have now generated a new construct of the wild-type (WT) TRAAK channel using the same backbone as that of the other mutants employed in the PDS experiments. WT TRAAK was measured under the same experimental conditions, as for all mutants described in our original manuscript. Furthermore, we remeasured all (including new) constructs using the voltage step protocol and assessed their sensitivity to arachidonic acid. The updated results are presented in **revised Figure S1** (reproduced as Figure 2 below), along with a statistical comparison of channel sensitivity to arachidonic acid and mechanosensitivity including wild-type (WT) TRAAK, InCys TRAAK, and E67C/L144C TRAAK, shown in Figure 2. These data support that none of the introduced mutations alter channel functionality. Additionally, we compared the currents of E67C/L144C TRAAK before and after MTSSL spin labelling (i.e., E67R1/L144R1) and observed no significant differences, indicating that spin labelling at positions 67 and 144 again does not have any effect on ion channel function (**panel b-g in the revised Figure 2**, reproduced as Figure 3 below).

Regarding temperature sensitivity, we recently purchased and installed a new temperature controller along with new high flow rate pumps integrated to our existing electrophysiology set up to enable stable and accurate current recordings at distinct temperatures over extended periods of time (i.e., up to one hour). Using our new setup, we tested 3 × replicates of WT TRAAK and observed no significant changes in channel activity when the temperature was increased from 24 °C to 41 °C under the inside-out patch clamp configuration. We also monitored channel activity continuously for 1 h at 37 °C (i.e. recordings above 37 °C were only possible for shorter time periods due to GUV patches being fragile and rupturing). Channel current did not increase with temperature, regardless of the incubation time (**revised Figure S13**, reproduced as Figure 4 below). Instead, total channel current gradually decreased over the course of the hour, which may be due to differences in GOhm seal formation over time at increasing temperatures. Our observation is consistent with previous studies, which showed that TRAAK activation upon heating only occurs in a whole-cell configuration but not in inside-out patches, suggesting that cell integrity is required for temperature activation of TRAAK in an electrophysiology set up (Kang et al., 2005; Maingret et al., 2000).

Figure 2: Comparison of channel activity for wild-type, InCys, and E67C/L144C TRAAK. a, g, Current responses to incremental negative pressure applied to representative excised

patches from wild-type (a) and InCys (g) TRAAK GUVs. Pressure steps every 5 s; traces vertically offset. Dashed red lines indicate the zero-current baseline. Holding potential (V_h) = 0 mV, holding pressure (Ph) = 0 mmHg. b, h, Currents recorded from patches of wild-type (b) and InCys (h) TRAAK during a voltage-step protocol ($V_h = -50$ mV; -100 to $+100$ mV in 10-mV increments; displayed every 40 mV). For each voltage step, a -50 mmHg pressure step was applied (lower). Dashed red line marks the current at $V_h = -50$ mV, $Ph = 0$ mmHg. c, I, I-V relationships from (b, h): average pre-pressure current and peak current during the pressure step vs voltage. d, j, m, Currents from excised patches of wild-type (d), InCys (j), and E67C/L144C (m) under a voltage-step protocol ($V_h = 0$ mV; -100 to $+100$ mV; $\Delta V = 10$ mV; displayed every 40 mV). e, k, n, Recordings from the same patches as (d, j, m) after 50 μ M arachidonic acid (AA) perfusion. f, l, o, I-V relationships corresponding to (d/e), (j/k), and (m/n), respectively. p, q, Fold change in current at $V_h = 0$ mV for wild-type, InCys, and E67C/L144C TRAAK in response to MS (-50 mmHg) and AA stimulation. MS increased current 1.66 ± 0.15 , 1.54 ± 0.10 , and 1.60 ± 0.04 -fold, respectively; AA increased current 1.47 ± 0.13 , 1.69 ± 0.06 , and 1.45 ± 0.14 -fold, respectively (mean \pm SEM). Biological replicate numbers (n) are shown above the bars. r, Fold change in current at $V_h = 0$ mV of wild-type TRAAK in response to lipid perfusion. TRAAK was activated 1.47 ± 0.13 , 1.58 ± 0.02 , and 1.51 ± 0.03 -fold by AA, PI, and PI(4,5)P2, respectively (mean \pm SEM, n = 3). In contrast, DOPE inhibited TRAAK by 2.00 ± 0.25 -fold (mean \pm SEM, n = 3).

Figure 3: b, Current response to pressure increments applied to patch excised from TRAAK E67C/L144C GUVs. Pressure steps were incremented every 5 sec, and the currents measured during each step are vertically offset for display. Dashed red lines below each current trace denote the zero current baseline. Holding potential (V_h) = 0 mV, holding pressure (Ph) = 0 mmHg. c, Recorded currents (Upper) obtained from a patch extracted from TRAAK E67C/L144C GUVs during a voltage step protocol ($V_h = -50$ mV, ranging from -100 to $+100$ mV, with a ΔV of 10 mV, displayed at intervals of 40 mV). For each voltage step, a pressure step of -50 mmHg was applied through the pipette (Lower). Dashed red line indicates the current level at the holding potential of -50 mV and holding pressure of 0 mmHg. d, The current-voltage relationship of the data presented in c is shown at 10 mV intervals. The average

current prior to pressure step and the peak current during the pressure step application are plotted against each voltage. **e, f**, Currents from excised patches of TRAAK E67C/L144C GUVs under a voltage step protocol ($V_h = 0$ mV; -100 to $+100$ mV in 10 mV increments; every 40 mV step shown), before (**e**) and after (**f**) perfusion with 1 μ M MTSSL. **g**, Current-voltage relationship from the data shown in **e** and **f**, incremented by 10 mV. The black trace represents the average current before MTSSL application, and the red trace (+MTSSL) represents the current after perfusion with 1 μ M MTSSL.

Figure 4: TRAAK is not activated by heating in the inside-out patch configuration. a, Currents from an excised inside-out patch of wild-type TRAAK recorded with a voltage-step protocol at ambient temperature (24 $^{\circ}$ C; $V_h=0$ mV; -100 to $+100$ mV; $\Delta V=10$ mV; displayed every 40 mV). **b, c**, Recordings from the same patch after the bath was raised to 37 $^{\circ}$ C (~ 15 min) and after 1 h at 37 $^{\circ}$ C, respectively. **d**, I–V relationships from the same patch under the three conditions.

3. While the PDS results clearly demonstrate that both the S and NS cap conformations coexist in TRAAK, and that their ratio can be modulated by temperature, the direct link between cap swapping and channel activity remains unclear. Specifically, it is not demonstrated that temperature-induced changes in cap conformation is related with an increase in current amplitude.

To further explore the functional relevance of cap swapping, the authors should consider testing gain-of-function mutations known to affect TRAAK gating as well as other stimuli such as

arachidonic acid. If such mutations and stimulus alter the S/NS equilibrium, as determined by PDS, and this correlates with changes in channel activity, it would provide strong evidence supporting the physiological relevance of the cap conformational dynamics. Moreover, to further investigate that NS to S conformational transitions are involved in TRAAK thermal sensitivity, it would be interesting to compare the S to NS ratios of TRAAK at different temperatures over the thermal range to which TRAAK is sensitive (from 30°C to 40°C).

Response:

We agree that establishing a direct link between cap swapping and channel activity is primarily limited by technical constraints, which was also the case for monitoring the entire cap ensemble of TRAAK in lipid membranes with high resolution prior to our study. First, as per our response above, activation of TRAAK by heating requires cell integrity (*Maingret, F. et al. EMBO J 19, 2483-2491, 2000; Kang, D. et al. J Physiol 564, 103-116, 2005*), which is absent in our current setup using reconstituted liposomes. Second, although PDS is a powerful technique that allows us to monitor the conformational landscape of the entire TRAAK population for the first time- including the minority-populated and so far neglected NS state- we are currently unable to separate the non-swapped (NS) from the swapped (S) TRAAK populations. Even if whole-cell patch-clamp recordings were available, we would still lack control over the exact proportion of TRAAK molecules present in each cap conformation within an individual recorded patch.

Thus, establishing a direct link between domain swapping and channel activity will require an additional breakthrough that enables separation of the two cap states. To note that even if currently available state-of-the-art antibodies are used to select for the S state in combination with PDS, still quantifying the entire ensemble will not be possible, while additionally using state-locking antibodies will not allow for unbiased monitoring of the entire ensemble. In this study, we provide a robust PDS-based method to detect and quantify the conformational states of TRAAK in the ensemble, representing an important first step toward investigating domain swapping in TRAAK, which could be also generalised for other K2P channels mechanosensitive or not (*TREK1/2, TWIK1, TASK et al*). One potential approach could be screening for a large number of conditions (and out of the scope of this study) using our current combined PDS and HSS-SL approach which may force the entire TRAAK population to a single species (i.e. ~100% S or NS). We would then be able to study the function (and structure) of this species separately and account for its individual contribution to TRAAK's activity.

Regarding arachidonic acid (AA) activation: We appreciate the valuable suggestion by the Reviewer and we have now performed additional PDS measurements in the presence of AA to examine whether there is a change in the S/NS equilibrium. However, the PDS data obtained from AA-treated spin labelled TRAAK samples were of insufficient quality (i.e. low SNR and very low EPR signal) to allow for reliable conclusions regarding its effect on the S/NS state. To investigate the origin of the poor EPR data quality after AA was added to heterologously spin-labelled TRAAK, we directly incubated the spin label MTSSL with AA and monitored the MTSSL signal by CW-EPR at room temperature and in solution. Surprisingly, we observed that AA progressively reduces the MTSSL signal over time until it completely diminishes after the course of few minutes (Figure 5).

Therefore, it seems that the poor quality of the PDS data and the substantially reduced cwEPR signal over AA incubation is due to the chemical properties of AA. Indeed, it has been reported that the bis-allylic groups in AA are prone to oxidation because their electrons are pulled by adjacent double bonds, weakening their bond energy (Johnson, D. R. et al. 2015). As a consequence, the diminishing EPR signal may reflect AA-induced reduction of the spin-label free radicals.

Figure 5: Time-dependent reduction of CW-EPR signal upon incubation of MTSSL with arachidonic acid (AA). (a) Continuous monitoring of the maximum intensity amplitude of the central line peak in the CW-EPR spectra during incubation of 17 μ M MTSSL with 6 mM AA shows a progressive decrease in signal over time. (b) Representative CW-EPR spectra from panel (a), recorded at three distinct time points, illustrate the decay in signal amplitude as incubation time increases.

Reference:

Johnson, D. R. & Decker, E. A. The Role of Oxygen in Lipid Oxidation Reactions: A Review. *Annual review of food science and technology*. 6, 171-190, doi:10.1146/annurev-food-022814-015532 (2015).

4. Authors carry out an analysis of the nature of lipids surrounding TRAAK. However, they do not clearly expose the aim of this analysis (beyond knowing the lipid partners of TRAAK, which of itself is interesting but constitutes an article on its own) and to what extent it is important within the framework of the article (swapped, non-swapped conformations). Furthermore, authors should provide more in-depth interpretation of their analysis.

Response:

We thank the Reviewer for the thoughtful comment. We now implemented additional electrophysiology experiments and analysis using the native lipids (i.e. identified by our ES-MS analysis) to account for their functional role. These data further support our conclusions and contribute to its completeness, as per Reviewer's suggestion to explore the functional role of the identified lipids. Our initial aim in analysing the immediate lipid environment of TRAAK

was to demonstrate the combined use of ES-MS with SMA detergent-free technology to study TRAAK in a native-like lipid environment.

Our ES-MS data revealed that TRAAK exhibits a preference for specific lipids which could have a functional role, an important aspect we now explore in our revised manuscript. To this end, we performed new multiple electrophysiology recordings of GUV-reconstituted TRAAK in presence of a) POPS, b) DOPE, c) PI(4,5)P₂, and d) PI lipids, identified in our previous ES-MS analysis to assess the functional effect each of these lipids have on TRAAK function.

As shown in **panel b-e from revised Figure 3**, (reproduced below as Figure 6), we found that in our set up both PI(4,5)P₂ and PI can activate TRAAK, while in contrast DOPE exerts an inhibitory effect. Finally, POPS does not significantly affect TRAAK's current amplitude, suggesting that this lipid likely plays a structural rather than regulatory role, possibly contributing to the stability of TRAAK within the lipid bilayer.

Fig. 6: TRAAK associates with specific membrane lipids and is modulated by lipid composition in reconstituted membranes. b–e, Representative currents from excised patches of wild-type TRAAK reconstituted in GUVs during a voltage-step protocol (V_h = 0 mV; steps from -100 to +100 mV in 10 mV increments; every 40 mV step shown).

show recordings before and after bath perfusion with 4 μM of the indicated lipid: PI (b), PIP₂ (c), DOPE (d), and POPS (e). Right panels: I–V relationships from the same patches at 10-mV increments. Black traces indicate average currents prior to lipid application, while red traces (+PI, +PIP₂, +DOPE and +POPS) show the response following lipid perfusion.

Minor points:

- “The equivalent of IPC in human membranes is sphingomyelin, which also has a choline headgroup, and would likely be undetected in our analysis.”

Why would it be undetected? The authors mention that PC lipids have previously been detected in similar experiments.

Response:

We thank the Reviewer for this observation. We cited several previous studies demonstrating that PC lipids can indeed be detected in SMA-derived lipid particles to clarify that the absence of PC in our analysis is not due to any known artifact or selective exclusion by the SMA polymer itself. Yeast do not make sphingomyelin, they make IPC instead. Therefore, the lack of PC detection in our study reflects a biological exclusion of PC from TRAAK's immediate lipid environment. This is particularly notable given that PC is the most abundant lipid species in *Pichia pastoris* membranes, which we used for protein expression. Based on this, we propose that sphingomyelin, another choline-containing lipid and the functional equivalent of IPC in human membranes, is also likely absent from the immediate environment of TRAAK and thus was not detected in our analysis.

- 1.359: “The initial and subsequent enhancement of this dominance of the S state within the TRAAK ensemble upon temperature increase, suggests the S state has greater thermodynamic stability than the NS state.” Shouldn't it be the reverse? Upon temperature increase, there is more energy in the system allowing more instable states to appear?

Response:

We thank the Reviewer for their comment and agree with their assessment regarding low energy states. This is indeed a quite critical point in our study, which we agree that needs clarification and requires further analysis. Because the S state is the dominant state at both low and high temperatures, we concluded that the S state has greater thermodynamic stability than the NS state.

However, to better understand the equilibrium shift caused by the higher temperature and address this specific point raised by the Reviewer, we teamed up with Dr Qaiser Waheed (now a new co-author in the revised manuscript and an expert in MD simulations) and performed long fully atomistic MD simulations of both the non-swapped (NS; PDB 3UM7) and swapped (S; PDB 4WFF) states at 19 °C and 40 °C. We then compared the non-bonded interaction energies between the protein and its entire environment (lipid bilayer and solvent). The results are shown in the manuscript as:

To better understand the shift of the equilibrium towards the S state, we performed independent atomistic molecular dynamics simulations of the NS state and the S state of TRAAK embedded in lipid bilayers at 19 °C and 40 °C and calculated the energies between the channel and its entire environment (Fig. S14a, Table S4). We first ensured systems in all four conditions are in equilibrium by monitoring the RMSD of the proteins through the trajectories (Fig. S14b). This approach enabled a direct comparison of the non-bonded interaction energy between the protein (accounting for its unique structural architecture) and its environment (lipid bilayer and solvent) at two distinct states and different temperatures. For the S state, the temperature dependence of the protein–environment interaction is relatively small ($\Delta\Delta G \approx 979 \pm 724$ kcal mol⁻¹) and more favourable at 19 °C. The NS state exhibits a larger energy difference ($\Delta\Delta G \approx 2689 \pm 819$ kcal mol⁻¹), also favouring 19 °C. Considering the fluctuations in the energy profiles and the resulting large errors (Fig. S14a, Table S4), S seems to be the most stable and the least temperature-sensitive state, whereas NS becomes less stable at elevated temperatures.

Discussion

The MD data and energy calculations are shown in **revised Figure S14** (reproduced below as Figure 7) and **revised Table S4** (reproduced below as Table 1).

Figure 7: Molecular dynamics energy calculations of the S and NS TRAAK states at two different temperatures. Time traces of (a) nonbonded interaction energies (Lennard–Jones and Coulomb electrostatic contributions) and (b) backbone RMSD between membrane-embedded TRAAK and its environment, compared across temperatures and channel states.

Table 1: Membrane–protein binding interaction energies for the S and NS states at two distinct temperatures. For each state, the energy difference between temperatures is also reported.

States	Temperature (°C)	Energy (kcal mol ⁻¹)	Energy difference (kcal mol ⁻¹)
S	19	-37821 ± 500.2	-979 ± 724
	40	-36842 ± 524.3	
NS	19	-42505.2 ± 548.9	-2689 ± 819
	40	-39816.2 ± 609.1	

- Authors speculate that either TRAAK adopts 2 distinct folding patterns during protein synthesis and/or that there is a conformational transition between the two states. But didn't they demonstrate that the latter occurs with their experiment on temperature? (Channels analyzed at low and high temperatures were synthesized with the same protocol.) Authors could mention it.

Response:

We thank the Reviewer for their insightful comment and for the opportunity to clarify point. While the temperature-dependent shift in the NS/S equilibrium suggests the presence of a conformational transition between these states, it does not exclude the possibility that TRAAK can adopt two distinct folding patterns during synthesis. The observation that both conformations exist prior to temperature treatment supports the idea that alternative folding pathways may also contribute to TRAAK's cap conformational example.

- 1.406: Authors provide an estimate of the swapped to non-swapped conformation transition energy (10 kcal/mol). They should compare it to the energy gain of the system when temperature is increased from 19°C to 40°C in their experiment. Additionally, they could provide a comparison with the energy associated to temperature shifts to which TRAAK is sensitive. This way, the proposition that a mechanism involving cap swapping explains TRAAK thermal sensitivity would be more convincing.

Response:

We thank the Reviewer for their valuable suggestion. Upon re-evaluating the available data, we realised that the previously reported estimate of ~10 kcal/mol for the cap region was obtained from simulations performed at 300 K (26.85 °C). This value reflects the energy difference for a limited number of residues within the cap region and does not represent the energetic profile of the full protein. Moreover, energy differences in protein conformational states are temperature-dependent, making it inappropriate to directly use this value to estimate the energy gain associated with increasing the temperature from 19 °C to 40 °C in our system.

Furthermore, it will not be fully accurate to directly relate the energy gain of the system to the energy required for the state transition. When the resuspended cell membrane carrying expressed TRAAK is heated in buffer solution, the absorbed energy is distributed (not evenly) into all (or multiple) samples components. Not all of the heat is retained by the system; part of it is dissipated, and part is transferred to other membrane proteins and lipids as well. Therefore, it is challenging to quantify the exact fraction of energy specifically transferred to TRAAK channels.

To better understand the PDS results which indeed show a shift of the equilibrium, we performed long atomistic simulations of both the non-swapped (NS; PDB 3UM7) and swapped (S; PDB 4WFF) states at 19 °C and 40 °C and compared the non-bonded interaction energy between the protein and its environment (lipid bilayer and solvent) (**revised Figure S14 and revised Table S4**)

(see also previous response). And the results are shown in the manuscript as:

To better understand the shift of the equilibrium towards the S state, we performed independent atomistic molecular dynamics simulations of the NS state and the S state of TRAAK embedded in lipid bilayers at 19 °C and 40 °C and calculated the energies between the channel and its entire environment (Fig. S14a, table S4). We first ensured systems in all four conditions are in equilibrium by monitoring the RMSD of the proteins through the trajectories (Fig. S14b). This approach enabled a direct comparison of the non-bonded interaction energy between the protein (accounting for its unique structural architecture) and its environment (lipid bilayer and solvent) at two distinct states and different temperatures. For the S state, the temperature dependence of the protein–environment interaction is relatively small ($\Delta\Delta G \approx -979 \pm 724 \text{ kcal mol}^{-1}$) and more favourable at 19 °C. The NS state exhibits a larger energy difference ($\Delta\Delta G \approx -2689 \pm 819 \text{ kcal mol}^{-1}$), also favouring 19 °C. Considering the fluctuations in the energy profiles and the resulting large errors (Fig. S14a, table S4), S seems to be the most stable and the least temperature-sensitive state, whereas NS becomes less stable at elevated temperatures.

- The paragraph on heterodimers might be over-speculated at this point, in particular regarding this sentence: “By quantifying the proportional representation of cap states within these ensembles, we can determine each state’s weighted contribution to the channel’s functional output.” Rephrase this sentence

We have now removed this sentence.

- In fig2.a, authors should mention the 2 mutations in the vector name: V5_GFP_TRAAK E67C L144C

We thank the Reviewer for spotting this mistake. Now corrected.

Mistakes or points of clarification in the text:

- 1.156: what do authors mean by “homologous TRAAK sites”?

We rephrase this sentence to:

For spin labelling of a single engineered site per subunit in homodimeric TRAAK, the unique architecture of the cap results in highly similar predicted distances for the two states

- 1.254: what does the “+ve” mean?

It means “positive ion mode” and it is corrected in the manuscript.

- 1.271: PIPS to PIPs

Corrected

- 1.316: contributed to contributing?

Corrected

Reviewer #4 (Remarks to the Author):

This manuscript probing the swapped (S) and non-swapped (NS) TRAAK potassium ion channel states using DEER spectroscopy is excellent. The research presented clearly indicates that both S and NS states coexist in membranes. The researchers noticed a shift in population to the swapped states at a higher temperature from 19 C to 40 C. The work presented in this study represents a clever way to get a complicated membrane protein sample that has a greatly increased DEER sensitivity for the state of the dimer. The DEER data shown in the paper is excellent. While a Tikhonov regularization is better in general for getting a sense of the most likely distance distribution, (there's no reason to assume that any distribution is purely Gaussian), using a dual gaussian fit is the best way to get a reasonably reliable measurement of the relative populations of the NS and S states (it's what everyone uses for

this). I think that this is compelling paper for a Nature Communications publication on the TRAAK K⁺ channel. Also, the methodology could be broadly applied to a lot of biological and membrane systems.

Response:

We sincerely appreciate the Reviewer's encouraging comments. We are pleased our combined methodology is praised by the Reviewer and particularly the fact they recognise the potential of a wider applicability of our integrative approach to other membrane protein systems.

Reviewer #5 (Remarks to the Author):

Potassium channels are oligomers of transmembrane subunits. The membrane-spanning domain of a multimeric channel is an assembly of alpha helices that interact with each other and with membrane bilayer lipids. The pore selectivity filter is formed by four reentrant domains which themselves interact with parts of the transmembrane helices. This assembly is highly stable - an essential condition for pore geometry and channel selectivity. Studies to date show that the oligomerization process takes place very early in channel biosynthesis in the endoplasmic reticulum, under the control of chaperone proteins. No studies have shown that the composition or overall conformational organization of the mature complex can be altered at the plasma membrane, probably because this would require too much energy. Under physiological conditions, very minor conformational changes modify the activity or selectivity of the channel and regulate its function.

Response:

We appreciate the thoughtful comments by the Reviewer. We agree that potassium channel assemblies comprising four pore-forming re-entrant domains are inherently highly stable structures and essential for maintaining optimal selectivity filter geometry. Consequently, domain swapping at the plasma membrane would require breaking strong inter-helical bonds which would be energetically unfavourable and likely destabilizing to the pore domain. However, while it is reasonable to question the likelihood of a large-scale conformational rearrangement after early biosynthesis in the ER, this does not entirely exclude the possibility of such phenomena occurring under certain conditions.

An illustrative example of post-folding assembly conformational rearrangement is found in the TRPV3 ion channel, which has been shown to transition from a tetrameric to a pentameric state in the membrane. Lansky et al., Nature, (2023) used high-speed atomic force microscopy (HS-AFM) to observe membrane-embedded TRPV3 and directly detected channels composed of five subunits, coexisting with the canonical tetramers. Importantly, these pentameric assemblies were transient and reversible, existing in dynamic equilibrium with tetramers through subunit exchange, with the entire process occurring within the membrane. HS-AFM videos captured real-time tetramer-to-pentamer and pentamer-to-tetramer transitions, with pentamers persisting for minutes within the lipid bilayer. The pentameric state was also trapped and resolved by CryoEM. This case study provides clear evidence of conformational variability in the mature TRPV3 complex, demonstrating that overall human ion channel organization can

be altered post biosynthetic assembly. Therefore, it remains possible that human TRAAK undergoes a conformational rearrangement at the membrane under certain physiological conditions.

Two-pore domain potassium (K2P) channels, including TRAAK, are dimers of subunits. Previous structural studies have shown that mature K2P channels can exist in two conformational states, the so-called non-swapped (NS) and swapped (S) forms. The two forms differ in the positions of the helices forming the extracellular cap and the membrane-spanning domains of each subunit in the dimer. The proportion of these two forms in cells and their functional significance are not known.

In this manuscript, Ma and colleagues expressed TRAAK in yeast, purified them and used pulse dipolar electron paramagnetic resonance spectrometry to calculate the proportion of N and NS forms. They also show that temperature influences this ratio suggesting that the previously reported effect of temperature on TRAAK could be due to a conversion between the N and NS forms. I'm not familiar with the technique used and the possible artifacts that it may produce, but the conclusions that TRAAK can switch between the N and NS states and that this switch has physiological significance are not supported by the provided data at this stage. Switching between the S and NS forms would require major conformational changes, with the unfolding and refolding of several domains and the modification of a considerable numbers of hydrophilic and hydrophobic interactions. No kinetics of this process are provided in the manuscript. If such rearrangement is possible after biosynthesis, then it is a multi-step process that should be slow. The effect of temperature on TRAAK/TREK1/TREK2 channel activity is rapid, as the effects of other stimuli (stretch, lipids, pH...). Also a 10-fold increase in TRAAK current is measured for a change from 17 to 40°C, while an increase of only 15% in the S form is observed. Nor is there any functional evidence that N and NS differ in their open probability or conductance, which would support the hypothesis of the authors.

Response:

We thank the Reviewer for their comment. We have now removed text which directly links TRAAK's activation upon temperature increase to domain swapping and state that heating alters NS/S equilibrium ratio. However, whether changes in the NS/S ratio contribute to the increase in current amplitude remains unknown. It is possible that two distinct mechanisms are involved.

In this work, the authors also took advantage of the way TRAAK is purified by SMA encapsulation to analyze the lipids surrounding the channel. While this is a promising approach, it has no bearing on the main message of the manuscript and its title. Much more work will be needed to show whether what is observed in yeast is relevant in mammalian cells for TRAAK function and/or whether other K2Ps behave differently or similarly to TRAAK with regard to the recruitment of surrounding lipids.

In conclusion, the authors have used novel and elegant approaches to study the structure of TRAAK and its lipid neighbourhood. But this work is still too preliminary to draw any firm conclusions about a functional role of the N and NS states of K2P channels.

We are pleased that the Reviewer finds our combined approach novel and innovative, but we disagree with their assessment that our study is preliminary. We have here successfully managed to unbiasedly monitor the entire conformational ensemble of TRAAK's cap domain in lipid membranes. However, we agree that what we observed in a yeast expression system will need to be tested in mammalian cell lines for TRAAK and other K2P channels. Such extensive experiments will require substantial time and resources, and we intend to explore in the future, but are well out of the scope of this study.

Nevertheless, we agree with the Reviewer that showing functional relevance of human-like (yeast) lipids identified in our ES-MS analysis will add to our current story. To this end we performed multiple additional electrophysiology experiments on TRAAK in presence of the equivalent (to yeast) human plasma cell membrane lipids identified in our ES-MS analysis. In brief (see also previous response to Reviewer 3), we performed additional electrophysiology recordings of GUV-reconstituted TRAAK in presence of a) POPS, b) DOPE, c) **PI(4,5)P₂**, and d) **PI** lipids, identified in our previous ES-MS analysis to assess the functional effect each of these lipids have on TRAAK function. We now show in **panel b-e from revised Figure. 3** (reproduced as Figure 6), that PI(4,5)P₂ and PI can activate TRAAK, while in contrast DOPE exerts an inhibitory effect. Moreover, we tested POPS, a lipid also found in our ES-MS native membrane analysis, which does not significantly affect TRAAK's current amplitude. These new electrophysiology data suggest that PS likely plays a structural rather than regulatory role for TRAAK, possibly contributing to the stability of the channel within the lipid bilayer.

Fig. 6: TRAAK associates with specific membrane lipids and is modulated by lipid composition in reconstituted membranes. b–e, Representative currents from excised patches of wild-type TRAAK reconstituted in GUVs during a voltage-step protocol ($V_h = 0$ mV; steps from -100 to $+100$ mV in 10 mV increments; every 40 mV step shown). Left and middle panels show recordings before and after bath perfusion with 4 μ M of the indicated lipid: PI (b), PIP₂ (c), DOPE (d), and POPS (e). Right panels: I–V relationships from the same patches at 10-mV increments. Black traces indicate average currents prior to lipid application, while red traces (+PI, +PIP₂, +DOPE and +POPS) show the response following lipid perfusion.

Response to comments

Reviewer #1 (Remarks to the Author):

I do not have any further comments for the authors. All my concerns have been addressed.

Reply: We thank the Reviewer for their constructive comments and are glad that all their concerns have now been addressed.

Reviewer #2 (Remarks to the Author):

Reviewer #3 (Remarks to the Author):

In their revised manuscript and their answers to the points we raised, Ma et al. have provided extensive supplementary data, additional details and justifications that strengthen their study on the temperature-dependent existence of the swapped and non-swapped conformations of the K2P TRAAK channel.

More specifically, authors successfully demonstrated through patch-clamp recordings that the mutated version of TRAAK used for their pulse dipolar EPR spectroscopy experiments is functional and retains the main regulatory properties of the wild type channel. We are now convinced of the scientific soundness of their experimental approach.

Regarding the part where authors study the lipids surrounding TRAAK, the authors clarified their analysis and answered our questions satisfactorily.

Finally, the authors successfully addressed our concerns regarding energetic considerations. The details they added in the manuscript as well as their molecular dynamics study largely improve the understanding of these energetic aspects.

However, even if authors clearly explain why it is technically challenging to assess the physiological relevance of the coexistence of the two TRAAK conformations (swapped and non-swapped), they failed to provide convincing evidence for the physiological significance of their findings.

Overall, their results represent a clear breakthrough in the field of K2P channel structural biology, and their approach is technically innovative. However, the limited physiological relevance of their findings may constrain the potential readership and overall impact of the study.

Reply: We thank the reviewer for recognizing the innovation and strengths of our study. We acknowledge that deep understanding of the physiological role of the coexistence of two cap conformations in TRAAK (and potentially other K2P channels) in lipid membranes requires further investigation, which is out of the scope of our current study. Although this aspect is technically challenging to investigate at present, it is an important direction that we plan to explore in our future work.

Reviewer #5 (Remarks to the Author):

We thank the authors for their response. As they agreed that "domain swapping at

the plasma membrane would require breaking strong inter-helical bonds which would be energetically unfavourable and likely destabilizing to the pore domain" and that it is therefore reasonable to "question the likelihood of a large-scale conformational rearrangement after early biosynthesis in the ER", it is important to mention this point in the manuscript as something to keep in mind for future studies.

Reply: We thank the Reviewer for this helpful suggestion. We have now incorporated this point in our revised manuscript.

A notable distinction between the S and NS states appears in the upper part of TRAAK's cap region, where both the orientation of Cys78 -essential for disulfide bond formation- and that of the three surrounding amino acids differ (Fig. 1a)²⁹. Since the two states co-exist, this could result from two possibilities; either TRAAK adopts two distinct folding patterns during protein synthesis and/or a conformational transition between the two states could occur. The latter would involve a 180-degree rotation of the opposing outer helices, which requires a large energetic penalty to break the existing or form new hydrogen bonds. How such a large-scale conformational rearrangement could occur after early biosynthesis in the endoplasmic reticulum requires further investigation.

Reviewer #6 (Remarks to the Author):

Reviewer #7 (Remarks to the Author):

This manuscript shows that swapped (S) and non-swapped (NS) TRAAK states co-exist in membranes and that their population shifts with temperature, supported by careful PDS/DEER measurements. The MD simulations aim to rationalize the temperature dependence by estimating "free energies" from non-bonded interaction energies. However, the current MD analysis has methodological and interpretational issues that require major revision.

Major Comments:

The statement that Coulombic interactions were "shifted to zero between 0–12 Å" implies that long-range electrostatics were truncated rather than treated with Ewald summation. For a membrane-protein system, this is not acceptable because electrostatics strongly govern protein–lipid and protein–solvent interactions. All simulations must be rerun with Particle Mesh Ewald (PME) to appropriately treat long-range interactions.

We thank the Reviewer for pointing this out to us. Our simulations were performed by combining the cut-off scheme with Force-switch for the Lenard-Jones, where the interactions are smoothly switched between rvdw switch at 1.0 nm and a cut-off at 1.2 nm. Coulomb interactions were calculated by using the cut-off type PME with a cut-off radius of 1.2 nm. Therefore, the interactions were calculated in real space within the cut-off, and in Fourier space using PME after the cutoff. We now include this information in the Methods section and clarify this point.

cell-rescale method (Bernetti and Bussi, 2020). Our simulations were performed by combining the cut-off scheme with Force-switch for the Lenard-Jones, where the interactions are smoothly switched between `rydwi` switch at 1.0 nm and a cut-off at 1.2 nm. Coulomb interactions were calculated by using the cut-off type PME with a cut-off radius of 1.2 nm. Therefore, the interactions were calculated in real space within the cut-off and in Fourier space using PME after the cutoff. The systems were subjected to production runs of 300–500 ns, using 2 fs time step. All simulations were performed using GROMACS¹¹⁰ versions 2021.5 and 2024.5 at the CSF3 Manchester and the local Pliotas Lab workstations accelerated by GPUs. Energy profiles describing non-bonded interactions between the protein and its environment (membrane and solvent) were computed for each state and temperature. Protein, membrane, and solvent were defined as separate energy groups, and the `gmx` energy tool in GROMACS was used to extract interaction energies. Non-bonded interactions included both Lennard–Jones (LJ) and Coulomb electrostatic (CE) contributions. The approximate interaction energy for each trajectory was estimated by considering the ideal gas as the reference state, as expressed in Eq. 1:

$$\Delta G = G(N, p, T) - G_{ideal\ gas}(N, p, T) = KT \ln(\langle e^{U/KT} \rangle) \quad (1)$$

The “approximate free energy” defined from total non-bonded interaction energy using an ideal-gas reference (Eq. 1) is unconventional and thermodynamically inconsistent. Entropic and PV terms are neglected, yet the quantity is interpreted as a free energy. Either rename this metric as an interaction-energy descriptor and avoid free-energy language, or replace it with a recognized method (alchemical FEP/TI, or a PMF along a physically motivated coordinate). If PV is omitted, provide quantitative justification.

We agree with the Reviewer, and we have now changed the terminology (i.e. to interaction energy) in the revised manuscript. We have further calculated the PV term and found that this is negligible compared to the high energetics contribution from the non-bonded interactions. This quantification analysis has now been included in the revised manuscript.

$$\Delta G = G(N, p, T) - G_{ideal\ gas}(N, p, T) = KT \ln(\langle e^{U/KT} \rangle) \quad (1)$$

where U is the sum of the LJ and CE interactions energies. PV was calculated by multiplying the reference pressure of 1 bar with the volume giving a contribution of 93 to 96 kJ/mol for all simulations. This contribution is negligible compared to the energies given in Table S4 and was therefore omitted. These calculated interactions represent the effective interaction energy of protein–membrane association. Due to long-timescale fluctuations in the interaction energies, strong energy drifts, a block-averaging scheme was applied in Eq. 1 to determine interaction energy values and associated errors. Differences in energies between simulations of the same state (with identical system composition) but at different temperatures were used to estimate the temperature dependence of the interaction energy change (Table S4). Analysis was performed for the last half of the trajectories through the python scripts using the `MAnalysis` package. This included the inner leaflet resolved 2D density map for the PS lipids headgroups normalized by the mole fraction of the bulk in the membrane plane, the radial distribution function between the PS lipids in the leaflet and the protein, area per lipid molecule in each leaflet and bilayer thickness. Three independent simulation replicates were performed and analysed, and all resulted in consistent outcomes.

The molecular origin of the temperature-dependent changes in non-bonded interactions remains unclear. Do these shifts primarily reflect membrane mechanics (e.g., changes in bilayer thickness, area per lipid, or leaflet asymmetry), or differences in the stability of the membrane-exposed cap? What is the pressure

profile along the membrane normal? Please quantify and discuss how these properties vary with temperature for both S and NS.

We thank the Reviewer for their thoughtful suggestions. We have now calculated these membrane properties, including membrane thickness and the area per lipid for each monolayer. These are now shown in Figure 1 below and the new Supplementary Figure S16 in the revised manuscript. Our new analysis reveals relatively small differences in membrane-thickness fluctuations.

For the NS state, the bilayer exhibits larger thickness fluctuations at 19 °C. These fluctuations reflect dynamic local deformations of the membrane around the protein. By allowing the bilayer to adapt more closely to the surface of NS-state TRAAK, these deformations lower the overall protein–membrane interaction energy, making the NS state more favourable at 19 °C. At 40 °C the membrane-thickness fluctuations are reduced, so the bilayer adapts less to the protein, leading to less favourable interactions and a higher interaction energy for the NS state. In contrast, for the S state the fluctuations in membrane thickness are highly similar for the two temperatures. These observations are consistent with our previous protein–environment interaction analysis, which showed that the temperature-dependent energy difference is larger for the NS state than the S state.

The number(s) of lipid molecules for the two leaflets show small asymmetries in PC (revised Supplementary Table S3). The NS state has four additional PC molecules in the outer leaflet. This explains the difference in area per lipid between the inner (cytoplasmic) and outer leaflets for the NS state, while the S state has one additional PC lipid molecule in the outer leaflet. The number(s) of PS and PE lipids are the same in each leaflet for the respective states, and the total number of lipids is constant across all trajectories.

NS 19 °C

NS 40 °C

S 19 °C

S 40 °C

Figure 1: Time evolution of membrane thickness and area per lipid for NS and S TRAAK simulations at different temperatures. In each row, the left panel shows the instantaneous bilayer thickness as a function of time; the red dashed lines and red numbers indicate the time-averaged thickness. The right panels show the corresponding area per lipid for the outer (gold) and inner (blue) leaflet of the bilayer, with horizontal dashed lines and coloured numbers indicating the respective time-averaged values.

The simulated bilayers contain DOPC:DOPE:POPS (72:14:14), i.e., an explicit fraction of phosphatidylserine (PS). Because PS is negatively charged and typically enriched in the inner leaflet, its spatial organization relative to TRAAK is crucial for interpreting the reported protein–membrane interaction energies. The manuscript should show where PS resides relative to the protein and whether it engages specific basic patches on the channel surface. Please provide leaflet-resolved 2D density maps of PS headgroups in the membrane plane, normalized by the bulk PS mole fraction, for both S and NS (with an inset radial profile $g(r)$ to quantify enrichment). Comparing these metrics between S and NS would clarify whether the reported energetic differences reflect PS-mediated contacts or generic bilayer mechanics.

In response to the reviewer’s request, we have computed leaflet-resolved two-dimensional density maps of PS headgroups in the inner leaflet, normalized by the bulk PS mole fraction, together with the corresponding radial distribution functions $g(r)$ around the TRAAK channel (Figure 2; new Supplementary Figure S17 in the revised manuscript).

For the NS state, the temperature dependence of the PS enrichment is more pronounced. At 19 °C, the 2D density map displays extended PS-rich patches around the protein, and the radial profile shows a clear maximum with $g(r)$ being significantly above a value of 2 at short distances away from the protein. At 40 °C, PS remains enriched relative to the bulk bilayer, but PS hotspots become weaker and more diffused, while the maximum of $g(r)$ is reduced. This indicates that PS molecules are more strongly and persistently associated with the NS state at 19 °C than at 40 °C.

For the S state, PS is also enriched near the channel, but the changes between 19 °C and 40 °C are comparatively modest: the $g(r)$ curves are more similar for the two temperatures, indicating that the PS environment of the S state is less temperature dependent.

To further address which specific regions of TRAAK interact with PS, we inspected the trajectories from all four conditions. In all cases, PS binding sites are mainly clustered in the channel regions between TM2–TM3 and TM1–TM4 (Figure 3). Because these interaction patterns are broadly similar across all conditions and do not highlight clear state- or temperature-specific differences, we did not include additional structural snapshots in the revised manuscript.

Importantly, all these new analyses and observations are consistent with our previous protein–environment interaction analysis. For the S state, the temperature dependence of the interaction energy was relatively small ($\Delta\Delta G \approx -979 \pm 724$ kcal mol⁻¹), whereas the NS state showed a significantly larger change ($\Delta\Delta G \approx -2689 \pm 819$ kcal mol⁻¹). Thus, the stronger temperature sensitivity of the NS state could arise from a combination of temperature-dependent PS (re)organisation around the channel and changes in local bilayer mechanics (e.g., membrane-thickness fluctuations), among other factors. A full dissection of all contributing terms would require a new set

of additional extensive simulations and analysis, which is beyond the scope of our present study.

Taken together, our new PS-density data are fully consistent with our previous analysis and further support the main conclusions of our original manuscript.

The change in the revised manuscript:

To further dissect the origin of the temperature-dependent interaction energies, we analysed individual membrane properties. For the NS state, the bilayer shows larger thickness fluctuations at 19 °C than at 40 °C (Fig. S16), indicating stronger local deformations that allow better adaptation to the protein and more favourable protein–membrane interactions at the lower temperature. In contrast, thickness fluctuations in the S state appear similar at both temperatures. We also examined the distribution of negatively charged PS lipids in the inner leaflet for all trajectories (Fig. S17). For the NS state, PS is enriched near the channel at 19 °C compared to 40 °C. This is evident in the extended PS-rich patches of the 2D maps and the higher value in the radial distribution function $g(r)$ of PS around the protein (Fig. S17). For the S state, the difference in $g(r)$ between the two temperatures is smaller, indicating that the PS environment is less dependent on temperature (Fig. S17).

Taken together, these results suggest that the stronger temperature sensitivity of the NS state arises from a combination of factors, including temperature-dependent PS (re)organisation around the channel, as well as changes in local bilayer mechanics.

NS 19 °C

NS 40 °C

S 19 °C

S 40 °C

Figure 2: 2D density maps of PS lipid headgroups and radial distribution functions for the inner leaflet of the NS and S TRAAK cap states at different temperatures. The left panels show the local PS headgroup mole fraction in the membrane plane (X–Y), normalised by the bulk PS mole fraction (colour bar; values > 1 indicate enrichment, values < 1 indicate depletion). The red contour indicates the projected outline of the protein complex. The right panels show the corresponding radial distribution function $g(r)$ of PS around the protein, with the dashed line marking $g(r) = 1$ (bulk density), quantifying PS enrichment or depletion in the inner leaflet.

Figure 3. Snapshots from each of the four simulation conditions, viewed from the cytoplasmic (inner leaflet) side. PS Lipids surrounding the protein (wheat-coloured sticks). In the NS state, the binding asymmetry is mostly confined to the structurally unresolved region between TM2 and TM3 of one of the two TRAAK subunits (pink coloured cartoon).

Minor Comments:

TRAAK gating is tension-sensitive. make the effective tension depend on leaflet lipid numbers and box-area fluctuations. Please clarify inner/outer leaflet lipid counts.

We now provide the total number of lipids in both leaflets in the revised Supplementary Table S3.

Structure	Lipid composition (DOPC:DOPE:POPS)	Total number of Lipids	Dimensions (X,Y,Z) (nm)	Temperature (°C)	Simulation length (ns)
3UM7	72:14:14	Upperleaflet:168	11.25,11.25,12.44	40	376
		Lowerleaflet:164	11.09,11.09,12.58	19	339
4WFF	72:14:14	Upperleaflet:165	11.18,11.18,12.70	40	498
		Lowerleaflet:164	11.03,11.03, 12.81	19	349

A representative snapshot showing the protein within the simulation box and the bilayer dimensions would help readers understand boundary conditions and system packing.

We now include a new Supplementary Figure S15 as follows:

Fig. S15: Representative snapshots showing NS and S TRAAK cap states within the bilayer of the simulation boxes. The water molecules are not shown for clarity. Grey spheres represent K^+ ions, and olive green spheres represent Cl^- ions. DOPC lipids are shown in wheat, POPS lipids in pink, and POPE lipids in cyan colours. Side views (top two panels) and cytoplasmic views (bottom two panels) are shown for clarity.